# A complex peripheral code for salt taste in *Drosophila*

Alexandria H Jaeger[1,2†], Molly Stanley[1†], Zachary F Weiss[1], Pierre-Yves Musso[1], Rachel CW Chan[3‡], Han Zhang[3], Damian Feldman-Kiss[1], Michael D Gordon[1]*

[1]Department of Zoology, University of British Columbia, Vancouver, Canada; [2]Graduate Program in Neuroscience, University of British Columbia, Vancouver, Canada; [3]Engineering Physics Program, University of British Columbia, Vancouver, Canada

**Abstract** Each taste modality is generally encoded by a single, molecularly defined, population of sensory cells. However, salt stimulates multiple taste pathways in mammals and insects, suggesting a more complex code for salt taste. Here, we examine salt coding in *Drosophila*. After creating a comprehensive molecular map comprised of five discrete sensory neuron classes across the fly labellum, we find that four are activated by salt: two exhibiting characteristics of 'low salt' cells, and two 'high salt' classes. Behaviorally, low salt attraction depends primarily on 'sweet' neurons, with additional input from neurons expressing the ionotropic receptor IR94e. High salt avoidance is mediated by 'bitter' neurons and a population of glutamatergic neurons expressing Ppk23. Interestingly, the impact of these glutamatergic neurons depends on prior salt consumption. These results support a complex model for salt coding in flies that combinatorially integrates inputs from across cell types to afford robust and flexible salt behaviors.
DOI: https://doi.org/10.7554/eLife.37167.001

*For correspondence:
gordon@zoology.ubc.ca

[†]These authors contributed equally to this work

Present address: [‡]Department of Computer Science, University of Toronto, Toronto, Canada

Competing interests: The authors declare that no competing interests exist.

## Introduction

Sodium is essential to survival, but its intake must be carefully regulated to maintain ionic homeostasis. It is therefore unsurprising that taste systems have evolved robust mechanisms for detecting salt, and that salt palatability depends on its concentration. In general, sodium concentrations below 100 mM tend to be attractive, while any salt present at higher concentrations becomes increasingly aversive (*Chandrashekar et al., 2010*; *Lindemann, 2001*; *Oka et al., 2013*)

Although there is considerable debate about modes of central taste coding, there is strong evidence that most taste modalities activate a single, molecularly defined, population of peripheral taste receptor cells (*Yarmolinsky et al., 2009*). However, research in both mammals and insects has favoured a dual-pathway model for salt taste: a low-threshold sodium-specific population of 'low salt' cells mediates attraction, which is overridden at higher concentrations by ion non-specific 'high salt' cells that drive avoidance (*Ishimoto and Tanimura, 2004*; *Lindemann, 2001*; *Marella et al., 2006*; *Oka et al., 2013*; *Zhang et al., 2013*). Moreover, two distinct aversive taste receptor cell (TRC) types (bitter and sour) contribute to high salt taste in mammals (*Oka et al., 2013*). Thus, peripheral coding of salt taste appears more complex than other primary taste modalities.

The *Drosophila* labellum contains three types of gustatory sensilla, each of which harbors 2–4 gustatory receptor neurons (GRNs) (*Singh, 1997*; *Stocker, 1994*)(*Figure 1A*). Short (S-type) and long (L-type) sensilla have four molecularly and physiologically distinct GRNs, while intermediate (I-type) sensilla have only two (*Freeman and Dahanukar, 2015*; *Scott, 2018*; *Stocker, 1994*). Extracellular 'tip-recordings' of different sensilla have identified four GRN types: a water (W) cell that responds to low osmolarity; a sugar (S) cell that responds to sweet compounds; a low salt (L1) cell that is sodium-specific; and a high salt (L2) cell that responds to high ionic concentrations (>250 mM) (*Fujishiro et al.,*

**eLife digest** Salt is essential for our survival, but too much can kill us. Our taste system has therefore evolved two different pathways to help us maintain balance. Low concentrations (like the salt on our chips) activate a pathway that makes us want to eat. But high concentrations (like the salt in seawater) activate pathways that do the opposite.

The nervous system takes on the role of detecting salt and encoding the information in a way that the brain can use. One specific type of cell detects each of the four other tastes: sweet, bitter, sour, and umami. But salt, with its two sensing pathways, is the exception to this rule. Previous work has examined salt taste responses in flies, but the picture is incomplete.

In flies, one type of taste neuron uses a different signaling mechanism to the others, suggesting that it might play a special role. So here, Jaeger, Stanley et al. asked how fly sensory cells encode salt information for the brain, and what those unusual neurons are for.

Mapping the taste receptor neurons in the tongue-like structure of the fly, the proboscis, revealed that salt information is not restricted to one or two types of cell. In fact, all five types of neurons tested (covering more than 90% of all the taste neurons present in flies) responded to salt in some way. Of these, two 'low salt' cell types made the fly want to eat salt, and two 'high salt' cell types made the fly want to avoid it. One of these high salt cell types was the unusual taste neuron identified previously. Rather than always encoding high salt as 'bad', the message from this type of cell changed depending on the diet of the fly. Salt-deprived flies ignored the activity of that cell type altogether. This complex way of encoding taste allowed the fly to change its behavior depending on how much salt it needed.

This work opens new questions, like how do the fly's neuronal circuits process this complex salt code? And how do the 'high salt' cells achieve their negative effect only when the need for salt is low? Understanding more about this system could lead to a better understanding of why our own brains enjoy salty foods so much.

DOI: https://doi.org/10.7554/eLife.37167.002

*1984*; *Hiroi et al., 2002*; *Ishimoto and Tanimura, 2004*). S- and L-type sensilla are thought to have one of each GRN type, with S-type L2 cells responding to bitter compounds in addition to high salt (*Meunier et al., 2003*)(*Figure 1B*). I-type sensilla were shown to have an S/L1 hybrid cell that responds to sugars and low salt, and an L2 cell that responds to bitters and high salt (*Hiroi et al., 2004*)(*Figure 1B*).

The early physiological recordings have been mostly borne out by molecular characterization of GRN types (*Freeman and Dahanukar, 2015*; *Scott, 2018*)(*Figure 1B*). S- and L-type sensilla each have a single GRN that expresses the low osmolarity sensor Pickpocket28 (Ppk28) and corresponds to the W cell (*Cameron et al., 2010*; *Chen et al., 2010*; *Inoshita and Tanimura, 2006*). The S cell is labelled by the sugar receptor Gr64f, along with other members of the gustatory receptor (GR) family (*Dahanukar et al., 2007*; *Fujii et al., 2015*; *Jiao et al., 2007*; *Slone et al., 2007*; *Thorne et al., 2004*; *Wang et al., 2004*). Similarly, Gr66a is co-expressed with other Grs in a single bitter responsive neuron per S-type and I-type sensillum, corresponding to the L2 cell (*Marella et al., 2006*; *Thorne et al., 2004*; *Wang et al., 2004*; *Weiss et al., 2011*). The degenerin/epithelial sodium channel (Deg/ENaC) family member Ppk23, which is required for pheromone detection in leg gustatory sensilla, is known to be expressed in a labellar neuron population that partially overlaps with Gr66a/ bitter GRNs (*Thistle et al., 2012*). Ppk23 neurons are necessary for calcium avoidance, but details of the labellar Ppk23 expression map, as well as the physiology and function of these neurons are largely unknown (*Lee et al., 2018*).

In contrast to water, sweet, and bitter tastes, the principles of peripheral salt coding in flies remain unclear. Early calcium imaging experiments revealed low salt responses in *Gr5a-Gal4* GRNs, suggesting that sweet neurons may mediate low salt attraction (*Marella et al., 2006*). However, *Gr5a-Gal4* was later shown to label additional GRNs outside the sweet class (*Fujii et al., 2015*), and the Ionotropic receptor (IR) family member IR76b was proposed to specifically mediate low salt taste via a dedicated low salt cell distinct from sweet GRNs (*Zhang et al., 2013*). This view was challenged by the recent demonstration that *IR76b* is also required for high salt taste, raising questions about

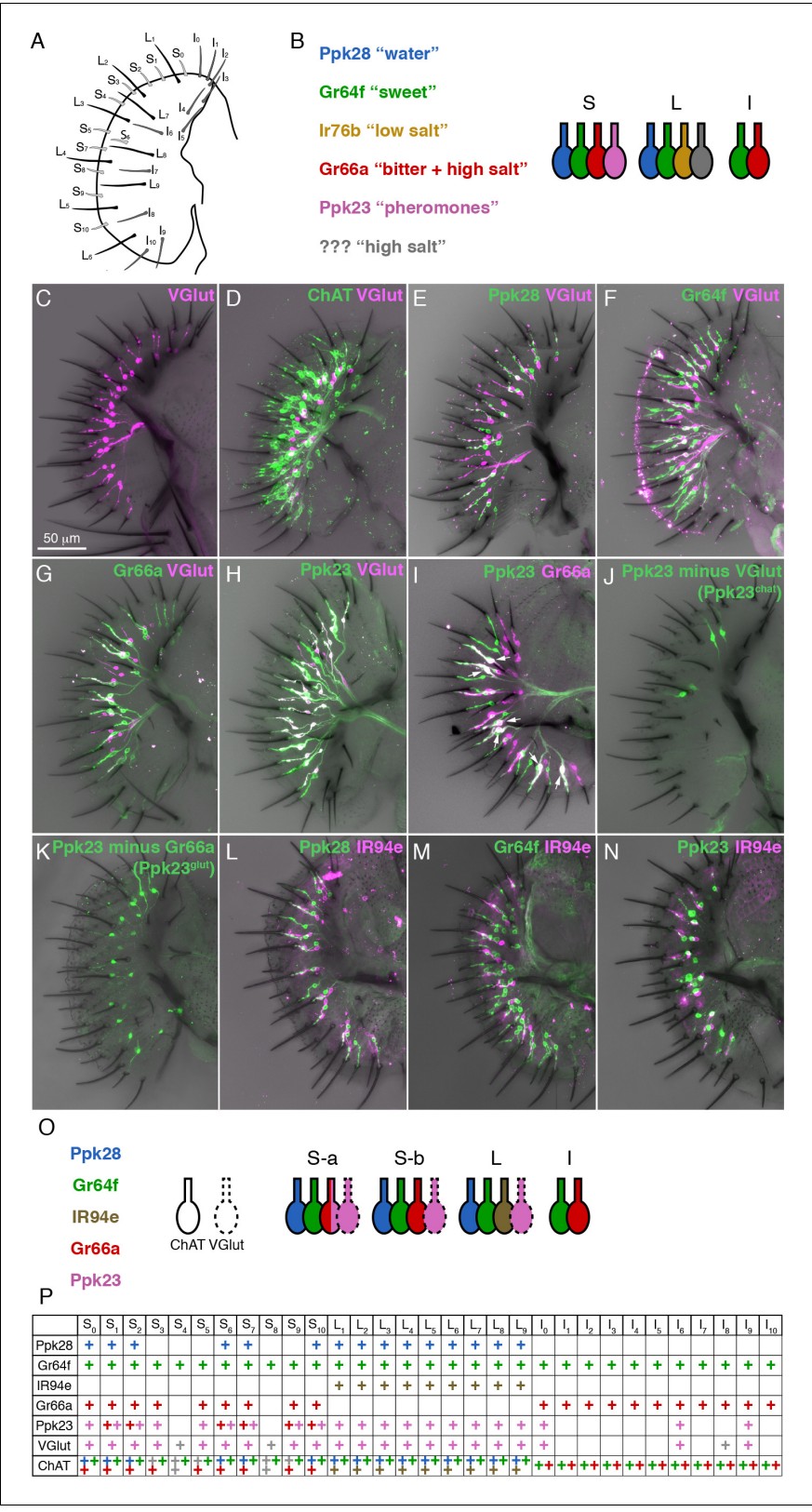

**Figure 1.** A molecular map of the fly labellum. (**A**) A schematic of sensillum identities in the fly labellum. (**B**) A summary of how GRN identities are currently viewed across the three sensillum types, with each color representing a GRN class with its most notable molecular label (if known) and its ascribed response properties. (**C–H**) Single labellar palps immunolabelled for *VGlut-Gal4* driving *UAS-tdTomato* (magenta) alone (**C**) or in combination with

*Figure 1 continued*

*LexAop-CD2::GFP* (green) under the control of *ChAT-LexA* (D), *Ppk28-LexA* (E), *Gr64f-LexA* (F), *Gr66a-LexA* (G), or *Ppk23-LexA* (H). (I) *Ppk23-LexA* (green) and *Gr66a-Gal4* (magenta) label partially overlapping populations. Arrows indicate sensilla where two Ppk23 GRNs exist, one of which co-expresses Gr66a. (J–K) Ppk23 subpopulations displayed by restricting *Ppk23-Gal4* expression with *VGlut-Gal80* (J), or *Gr66a-LexA* and *LexAop-Gal80* (K). (L–N) *IR94e-Gal4* labellum expression (magenta), co-labelled with *Ppk28-LexA* (L), *Gr64f-LexA* (M), and *Ppk23-LexA* (N). (O) Summary of newly defined GRN types following our mapping experiments. (P) Detailed map of each sensillum. Colors of '+' in chart indicate cell type. Grey denotes unknown identity. The VGlut +GRNs observed in I-type sensilla were sporadic and small, which is why they are not considered in the summary.

DOI: https://doi.org/10.7554/eLife.37167.003

The following figure supplement is available for figure 1:

**Figure supplement 1.** Molecular mapping of characterized and uncharacterized labellar GRN types.
DOI: https://doi.org/10.7554/eLife.37167.004

its utility as a marker for one defined GRN population (*Lee et al., 2017*). Moreover, although Gr66a GRNs showed calcium responses to high salt concentrations and electrophysiology suggested that bitter and high salt are encoded by the same sensory neurons, genetically eliminating these cells left behavioral aversion to high salt largely intact (*Marella et al., 2006*; *Wang et al., 2004*).

Here, we probe the logic of salt coding across the labellum by systematically characterizing the physiological and behavioral roles of molecularly-defined GRN types covering the entire labellar GRN map. We find that all GRN types show dose-dependent excitation or inhibition by salt, indicating a complex model for salt coding. Of particular interest is that, like mammals, flies have two distinct high salt cells. In addition to activating canonical bitter neurons, high salt concentrations excite a glutamatergic GRN population expressing Ppk23. Salt responses of these 'Ppk23$^{glut}$' GRNs require IR76b, whereas those of bitter GRNs do not. Both bitter and Ppk23$^{glut}$ GRNs are necessary for behavioral avoidance of high salt when flies have been reared on a salt-containing diet. However, salt deprivation reduces high salt avoidance by specifically suppressing the impact of Ppk23$^{glut}$ neurons, suggesting that these GRNs mediate internal state-dependent modulation of salt consumption. Consistent with this idea, closed-loop optogenetic activation of Ppk23$^{glut}$ neurons reduces feeding by salt-fed flies, but not those that have been salt deprived. Our results support a model where the combinatorial excitation and inhibition of various taste pathways mediates the behavioral valence of salt, with one pathway conferring the ability to specifically modulate salt consumption based on internal state.

## Results

### A comprehensive map of GRN classes in the labellum

Although several studies have mapped the expression of different receptors across the labellum (*Freeman and Dahanukar, 2015*), a comprehensive map covering all GRN types was still lacking. We began by asking whether the vesicular glutamate transporter (VGlut) may define a functionally distinct population of GRNs. An enhancer trap upstream of *VGlut*, *OK371-Gal4*, labels an uncharacterized population of putatively glutamatergic neurons in the labellum, and a subset of pheromone-responsive GRNs in the legs (*Kallman et al., 2015*; *Mahr and Aberle, 2006*). *VGlut$^{MI04979}$-Gal4*, which is a gene-trap inserted within a *VGlut* exon, showed expression in a single cell per S-type and L-type sensillum in the labellum (*Figure 1C*). These cells do not overlap with those expressing a similar gene trap for choline acetyltransferase (ChAT), supporting the idea that *VGlut$^{MI04979}$-Gal4* labels a bona fide population of glutamatergic GRNs (*Figure 1D*).

Co-labelling of *VGlut$^{MI04979}$-Gal4* with *LexA* reporters for known sensory neuron populations revealed that VGlut is not expressed in water (Ppk28), sweet (Gr64f), or bitter (Gr66a) GRNs (*Figure 1E–G*). However, all VGlut$^+$ cells were positive for Ppk23 (*Figure 1H*). Further examination of Ppk23 expression revealed that Ppk23 GRNs are comprised of two distinct subsets: most S-type and all L-type sensilla contain a single GRN that expresses Ppk23 and VGlut; and six S-type sensilla, roughly corresponding to those designated as 'S-a' sensilla (*Freeman and Dahanukar, 2015*; *Weiss et al., 2011*), have a second Ppk23 GRN that is positive for Gr66a and ChAT (*Figure 1I*). We

will refer to these two populations as Ppk23[glut] and Ppk23[chat], respectively. We then used gene trap insertions of *Gal80* into *VGlut* and *ChAT* to isolate Ppk23[chat] and Ppk23[glut] GRNs. While this restriction largely agreed with our co-expression data, it was not perfect: *VGlut-Gal80* restricted *Ppk23-Gal4* expression to four S-type GRNs instead of the expected six (*Figure 1J*); and *ChAT-Gal80* suppressed *Ppk23-Gal4* expression in most, but not all, Ppk23[chat] GRNs, leaving 1 – 2 s-type sensilla with two Ppk23 neurons (*Figure 1—figure supplement 1A*). We suspect these results reflect minor differences between expression of the *Gal4, LexA,* and *Gal80* reporters, although we confirmed that *Ppk23-Gal4* and *Ppk23-LexA* labelled the same population of GRNs (*Figure 1—figure supplement 1B*). To create a more conservative representation of Ppk23[glut], we constrained *Ppk23-Gal4* activity using *Gr66a-LexA* and *LexAop-Gal80* (*Figure 1K*). This manipulation faithfully restricted expression to only Ppk23[glut] GRNs, but also globally reduced expression levels. Thus, we retained both methods of isolating Ppk23[glut] cells for functional characterization.

Our analysis of Ppk23 expression nearly completed the labellar GRN map: s-type sensilla generally have one Ppk28 (water), one Gr64f (sweet), one Gr66a (bitter, some of which are Ppk23[chat]), and one Ppk23[glut] GRN; I-type sensilla have one Gr64f and one Gr66a GRN; and L-type sensilla have one Ppk28, one Gr64f, one Ppk23[glut], and one unidentified GRN that has been proposed to express IR76b and respond to low salt concentrations (*Freeman and Dahanukar, 2015*; *Zhang et al., 2013*). To identify a marker for the last GRN class in L-type sensilla, we first examined IR76b. However, *IR76b-Gal4* is expressed in many neurons from all four known classes of labellar GRNs, limiting its utility as a marker (*Figure 1—figure supplement 1C–F*). We therefore visually screened the Vienna Tile (VT) and Janelia Rubin Gal4 collections for lines that sparsely label GRN projections in the brain, and identified *VT046252-Gal4*, which drives Gal4 expression under the control of the genomic region upstream of the *IR94e* locus. Because the labellar projections to the subesophageal zone (SEZ) labeled by *VT046252-Gal4* (*Figure 1—figure supplement 1G*) appear identical to those of a previously published reporter for IR94e expression (*Koh et al., 2014*), we will henceforth simply refer to it as *IR94e-Gal4. IR94e-Gal4* is expressed in one cell per L-type sensillum, and does not overlap with Ppk28, Gr64f, or Ppk23 (*Figure 1L–N*). This driver is therefore specific for the fourth GRN class found in L-type sensilla and completes our molecular map of the labellum (*Figure 1O–P*).

## All labellar GRN classes respond to salt

With a complete labellar GRN map in hand, we examined the salt responses across all identified GRN classes. We expressed GCaMP6f under the control of each GRN class-specific Gal4 line and performed imaging of GRN axon terminals in the SEZ while stimulating the labellum with a series of tastants (*Figure 2A–D*). As expected, known GRN classes responded strongly to their cognate modality – Ppk28 to water, Gr64f to sugar (sucrose), and Gr66a to bitter (lobeline) (*Figure 2C–D*). As previously demonstrated, Ppk28 neurons show dose-dependent inhibition by salt, as with any osmolyte (*Cameron et al., 2010*). In contrast, Gr64f and Gr66a both showed dose-dependent excitation by salt, with Gr64f GRNs activated at a lower threshold. Moreover, Gr64f responses were sodium-specific, while Gr66a also responded to potassium chloride (*Figure 2C–D*). These results are consistent with Gr64f operating as a 'low salt' cell type, and Gr66a acting as a 'high salt' cell type.

Strikingly, we found that the two relatively uncharacterized labellar GRN types – IR94e and Ppk23 – also showed salt-evoked activity (*Figure 2C–D*). IR94e displayed weak activation by 50 mM NaCl, but no responses to higher concentrations. Further testing of different salts at 100 mM revealed sodium-selective tuning, indicating that IR94e labels a second low salt cell type (*Figure 2—figure supplement 1A–B*). The weak responses in IR94e neurons suggest a limited role in salt coding, but it is possible that they account for the previously observed peak response to low salt in L-type sensilla (*Zhang et al., 2013*). On the other hand, Ppk23 neurons showed very strong dose-dependent salt responses that were ion non-selective. In addition to sodium chloride, we observed robust activation by 1 M solutions of potassium chloride, sodium bromide, potassium bromide, cesium chloride, and calcium chloride (*Figure 2—figure supplement 2A–B*). As confirmation that the observed activity is salt-evoked and not a response to high osmolality, we found that Ppk23 GRNs do not respond to 1 M concentrations of sucrose (*Figure 2—figure supplement 2A–B*).

Since Ppk23 neurons on the leg are known to sense pheromones, we also tested labellar Ppk23 GRN responses to male and female cuticular hydrocarbons. We observed only very weak activation of Ppk23 neurons in female flies to a mixture of two male pheromones, and no significant responses in male flies (*Figure 2—figure supplement 2C–D*). Together, these data suggest that a primary

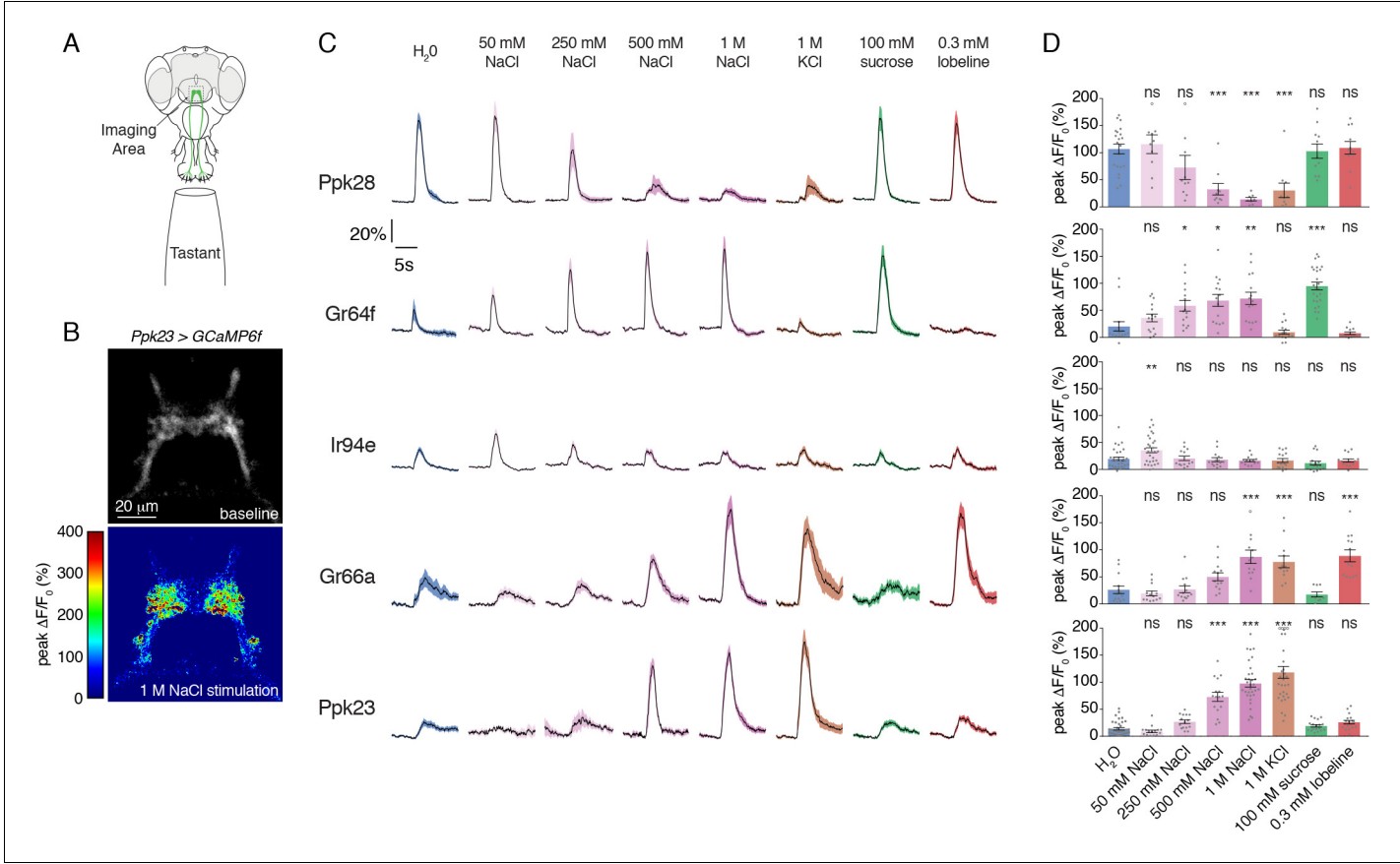

**Figure 2.** Salt activates or inhibits every GRN class. (A) Schematic of calcium imaging preparation. Taste neurons are stimulated on the proboscis, while GCaMP6f fluorescence is recorded at the synaptic terminals in the SEZ. (B) Representative heat map showing activation of Ppk23 GRNs with 1 M NaCl. (C) GCaMP6f fluorescence changes over time, following stimulation of each GRN class with the indicated tastants. Lines and shaded regions represent mean ±SEM, with stimulation occurring at 5 s. In each case, *UAS-GCaMP6f* is expressed under the control of the indicated *GRN-Gal4*, with the exception of Ppk28, which is from *Ppk28-LexA* and *LexAop-GCaMP6f*. (D) Peak fluorescence changes during each stimulation. Bars represent mean ±SEM. n = 8 – 37. Open circles indicate values that were higher than the y-axis maximum. Asterisks indicate significant difference from water by one-way ANOVA with Bonferroni post hoc test, *p<0.05, **p<0.01, ***p<0.001.

DOI: https://doi.org/10.7554/eLife.37167.005

The following source data and figure supplements are available for figure 2:

**Source data 1.** Raw numerical data for *Figure 2* and associated figure supplements.
DOI: https://doi.org/10.7554/eLife.37167.008
**Figure supplement 1.** IR94e neurons show weak, low sodium-specific responses to salt.
DOI: https://doi.org/10.7554/eLife.37167.006
**Figure supplement 2.** Ppk23 neurons respond strongly to all salts and only weakly to pheromones.
DOI: https://doi.org/10.7554/eLife.37167.007

function of labellar Ppk23 GRNs is to mediate a high salt response, and position Ppk23 and Gr66a as markers of two high salt GRN classes.

## Functional subdivision of Ppk23 GRNs

Our expression mapping revealed that Ppk23 GRNs encompass two subsets based on neurotransmitter expression: Ppk23[chat] and Ppk23[glut]. Given that Ppk23[chat] GRNs also express Gr66a, we suspected that this subpopulation may confer bitter responses to the Ppk23 population when measured as a whole. Although we did not observe Ppk23 activation in response to 0.3 mM lobeline (*Figure 2C–D*), we did see strong responses to caffeine (*Figure 3A*). Interestingly, we observed a marked difference in the synaptic calcium signals in response to salt and bitter stimuli. While salt stimulation of Ppk23 GRNs resulted in predominantly lateral activation of Ppk23 projections, bitter

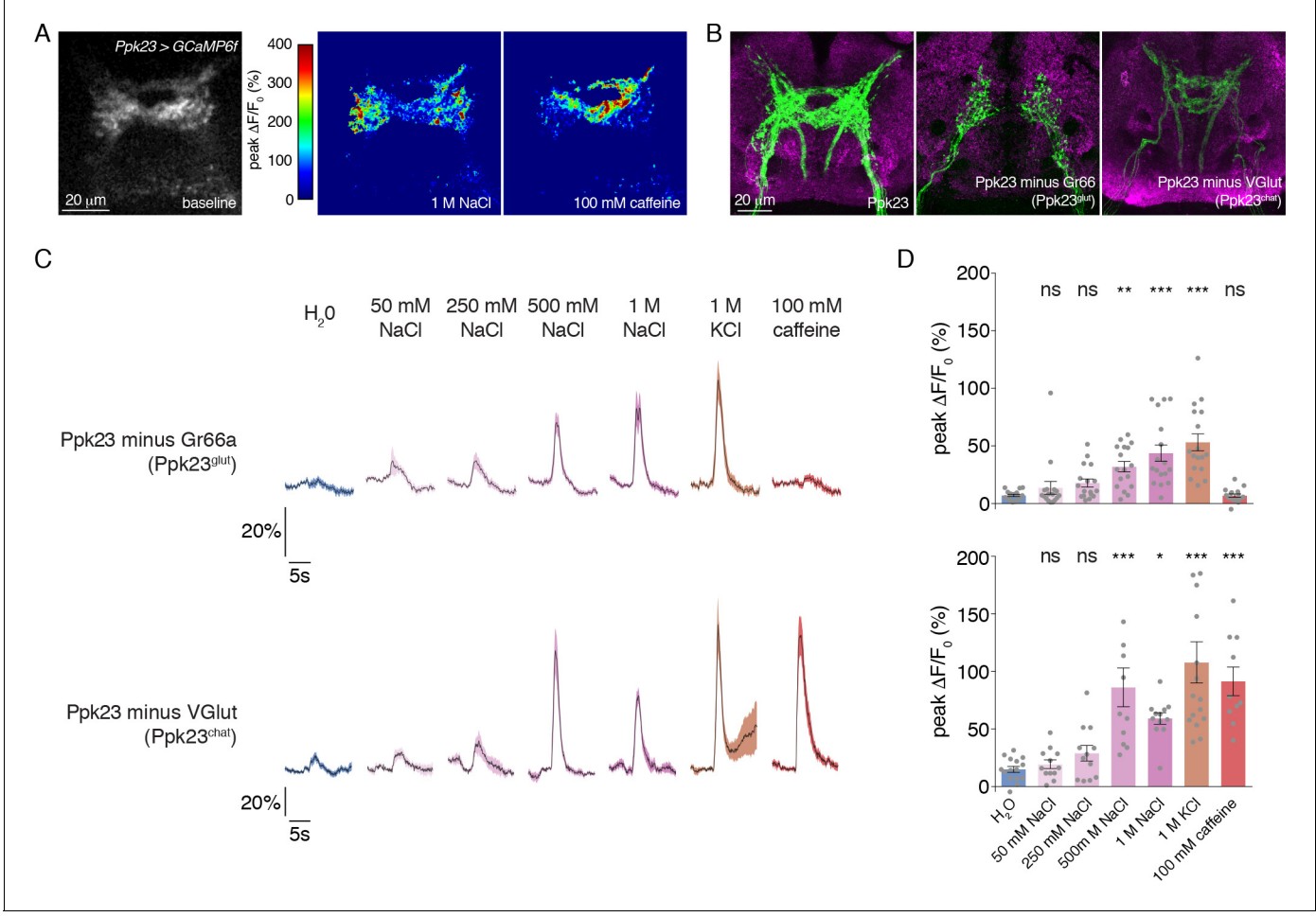

**Figure 3.** Morphological and functional distinction between Ppk23 subclasses. (A) Representative heat maps showing the activation Ppk23 GRNs in a single fly stimulated with 1 M NaCl and 100 mM caffeine. Salt primarily results in lateral activation, while bitter activates medial projections. (B) The projections of Ppk23 subsets targeting the SEZ, as revealed by immunofluorescent detection of GFP (green). Neuropil is labeled by nc82 (magenta). The full Ppk23 population targets both medial and lateral regions (left panel). Ppk23[glut] GRNs, revealed by restriction of *Ppk23-Gal4* with *Gr66a-LexA* and *LexAop-Gal80*, target only lateral areas (middle panel). Ppk23[chat] projections, revealed by restriction of *Ppk23-Gal4* with *VGlut-Gal80*, project to medial targets (right panel). (C) GCaMP6f fluorescence changes over time, following stimulation of each Ppk23 subset with the indicated tastants. Each fly has *Ppk23-Gal4* driving *UAS-GCaMP6f*, restricted by either *Gr66a-LexA* and *LexAop-Gal80* (top row) or *Vglut-Gal80* (bottom row). Lines and shaded regions represent mean ±SEM, with stimulation occurring at 5 s. (D) Peak fluorescence changes during each stimulation. Bars represent mean ±SEM, n = 10–17. Asterisks indicate significant difference from water by one-way ANOVA with Bonferroni post hoc test, *p<0.05, **p<0.01, ***p<0.001.

DOI: https://doi.org/10.7554/eLife.37167.009

The following source data and figure supplement are available for figure 3:

**Source data 1.** Raw numerical data for *Figure 3* and associated figure supplements.
DOI: https://doi.org/10.7554/eLife.37167.011

**Figure supplement 1.** Ppk23[glut] GRNs respond to high salt.
DOI: https://doi.org/10.7554/eLife.37167.010

stimulation activated medial ring-like projections characteristic of Gr66a (*Figure 3A*)(*Kwon et al., 2014*; *Thorne et al., 2004*; *Wang et al., 2004*). These activation patterns matched closely with the projections of the Ppk23[glut] and Ppk23[chat] subsets, revealed by restricting *Ppk23-Gal4* activity with *Gr66a-LexA* and *LexAop-Gal80* (Ppk23[glut]) or *VGlut-Gal80* (Ppk23[chat])(*Figure 3B*). This suggests that Ppk23[glut] and Ppk23[chat] both respond to salt, but that only Ppk23[chat] responds to bitter compounds.

To confirm this and reveal any functional differences in salt coding between the Ppk23 subpopulations, we measured the tuning of Ppk23[glut] and Ppk23[chat] using calcium imaging. As expected,

Ppk23[chat], but not Ppk23[glut], GRNs exhibited bitter responses; however, both subpopulations showed strong dose-dependent excitation by salt (*Figure 3C–D*). The salt responses in Ppk23[glut] GRNs appeared smaller than those of Ppk23[chat], but we suspected this was due to very low GCaMP6f expression in Ppk23[glut], which reduced the signal-to-noise in those measurements. We therefore repeated the Ppk23[glut] imaging by restricting *Ppk23-Gal4* expression with *ChAT-Gal80*. Consistent with the imperfect restriction we observed in the labellum (*Figure 1—figure supplement 1A*), these flies had small, although insignificant, caffeine responses (*Figure 3—figure supplement 1A–B*). However, this primarily Ppk23[glut] population exhibited very strong activation by salt (*Figure 3—figure supplement 1A–B*).

Taken together, our anatomical and functional studies support a salt coding model with two functionally distinct high salt GRN populations: Gr66a and Ppk23[glut]. We currently lack evidence of any functional distinctions between Gr66a GRNs that are positive or negative for Ppk23. Therefore, for the purposes of salt coding, we will consider Gr66a GRNs as a uniform population that includes Ppk23[chat].

## IR76b is necessary for low and high salt responses

A previous report suggested that IR76b is specifically required for low salt responses in an L-type GRN class distinct from Gr64f (*Zhang et al., 2013*). However, more recent evidence points to a role in both high and low salt taste (*Lee et al., 2017*). Since we observed widespread *IR76b-Gal4* expression in many GRN classes, we sought clarity on the role of *IR76b* in salt taste responses across the labellum.

Calcium imaging in *IR76b* mutants revealed that IR76b is absolutely required for salt-evoked activity in Gr64f GRNs (*Figure 4A–B*). By contrast, the salt responses of Gr66a GRNs were only mildly decreased in the mutants, showing that these neurons have a mostly IR76b-independent mechanism for detecting high salt. Ppk23 salt responses had a much stronger dependence on IR76b, with significantly decreased peak values, compared to controls, at all concentrations tested (*Figure 4—figure supplement 1A–B*).

Given the IR76b-independent salt responses in Gr66a GRNs, it was unsurprising that *IR76b* mutants showed some Ppk23 GRN activity in the medial region targeted by Ppk23[chat] (Gr66a-positive) projections (*Figure 4C*). We therefore reanalyzed the Ppk23 dataset by quantifying fluorescence change in a region-of-interest restricted to the lateral areas characteristic of Ppk23[glut] projections (*Figures 3B* and *4C*). *IR76b* mutants exhibited essentially no salt-evoked activity in this target region, suggesting that IR76b is necessary for both the sodium and potassium salt responses of the Ppk23[glut] population (*Figure 4A–B*).

Since *IR25a* is expressed in GRNs and thought to be another broadly acting co-receptor (*Ahn et al., 2017a*; *Benton et al., 2009*; *Cameron et al., 2010*; *Chen and Amrein, 2017*; *Lee et al., 2018*), we also tested its involvement in salt taste. We found that *IR25a* mutants have GRN response profiles similar to those of *IR76b* mutants, suggesting that perhaps IR25a and IR76b act in a complex to mediate gustatory salt responses (*Figure 4—figure supplement 2*). However, in contrast to *IR76b* and *IR25a*, mutations in *Ppk23* and the related ENaC *Ppk29* had no observable effect on the salt-evoked calcium responses of Ppk23 GRNs, consistent with previously reported behavioral tests (*Figure 4—figure supplement 3A–B*, [*Thistle et al., 2012*]). Thus, the *Ppk23* gene marks a salt-responsive GRN population but does not appear to be involved in salt detection.

The fact that *IR76b* mutants lack salt responses in the primary low salt GRN class and one of two high salt GRN classes provides an explanation for observed defects in both low salt attraction and high salt avoidance (*Lee et al., 2017*; *Zhang et al., 2013*). Before further dissecting the cellular contributions of different GRN classes to salt behaviors, we wanted to establish behavioral assays that replicated these phenotypes. To test low salt attraction, we used a binary choice assay where flies were given the option to feed on either 50 mM salt mixed with low sugar (2 mM sucrose), or the same concentration of sugar alone (*LeDue et al., 2015*; *Tanimura et al., 1982*; *Zhang et al., 2013*). As previously reported, control flies are strongly attracted to the salt-containing option, while *IR76b* mutants lose this attraction (*Figure 4D*). We used a similar assay to probe high salt avoidance. In this case, control flies avoid 250 mM salt mixed with 25 mM sucrose in favor of plain sucrose at a lower concentration (5 mM). Much like their defects in low salt attraction, *IR76b* mutants are severely impaired in high salt aversion (*Figure 4E*).

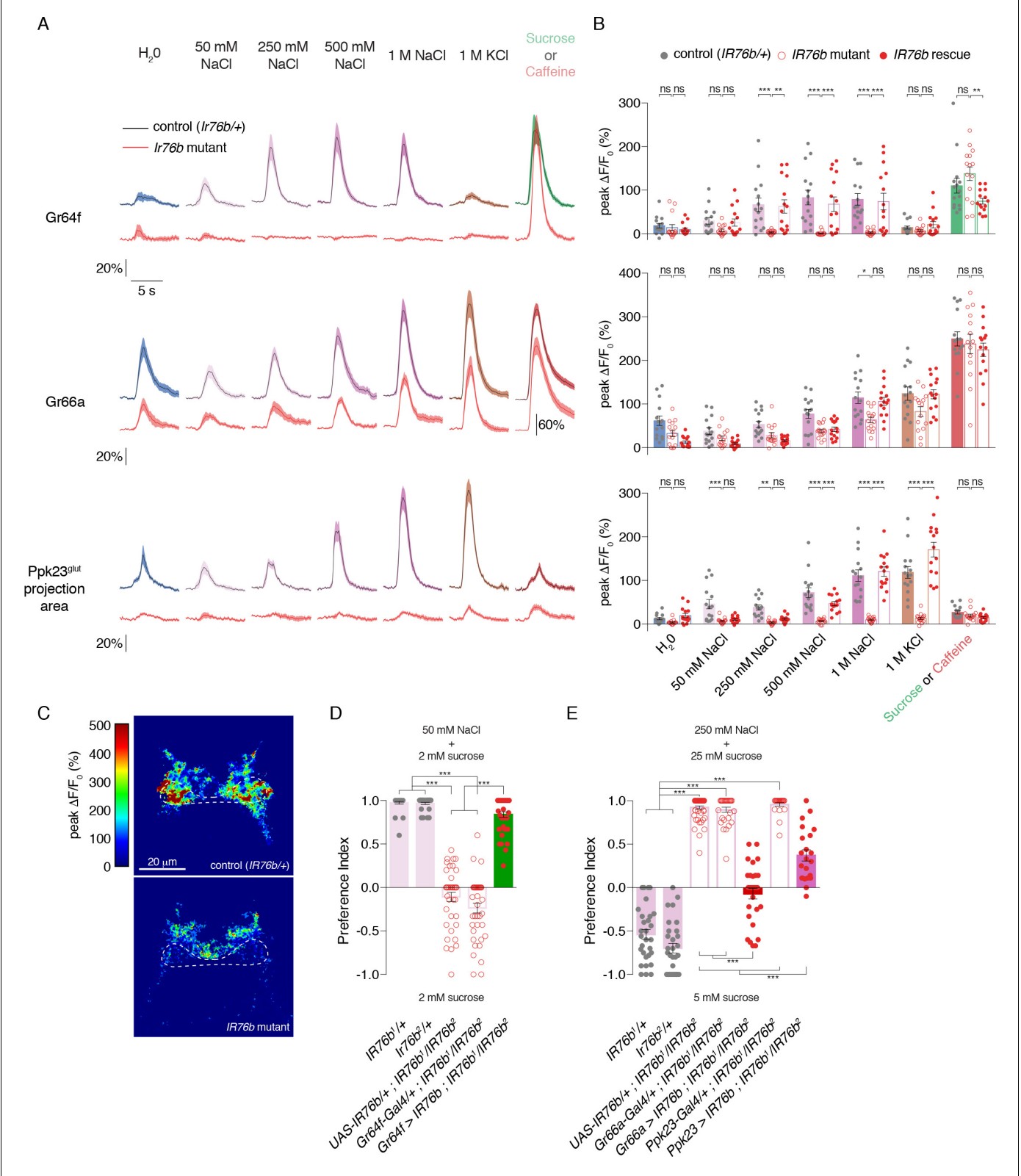

**Figure 4.** IR76b is necessary for Gr64f and Ppk23$^{glut}$, but not Gr66a, salt responses. (**A**) GCaMP6f fluorescence changes over time for each indicated GRN type, following stimulation with the denoted tastants. Black lines are for control genotypes (*IR76b$^1$/+* background), red lines for *IR76b$^1$/IR76b$^2$* mutants. '20%' scale bar refers to all curves except for the Gr66a caffeine curves, which are scaled by the '60%' bar. For Ppk23$^{glut}$, GCaMP6f was expressed under the control of *Ppk23-LexA*, but only the lateral regions corresponding to Ppk23$^{glut}$ projections were quantified. The pre-stimulus and

*Figure 4 continued on next page*

*Figure 4 continued*

post-stimulus periods were truncated in all curves to more clearly illustrate stimulus phase. (B) Peak fluorescence changes during each stimulation. Bars represent mean ±SEM, n = 15 for each stimulus. Filled grey circles (control), open red circles (*IR76b* mutants), and filled red circles (cell-specific *IR76b* rescue in imaged GRN population) indicate values for individual replicates. Asterisks indicate significant difference between control and mutant or mutant and rescue responses for each stimulus by two-way ANOVA with Bonferroni post hoc test, $*p<0.05$, $**p<0.01$, $***p<0.001$. (C) Representative heat maps of salt-evoked activity in Ppk23 neurons from control (top) and *IR76b* mutant (bottom) animals. Activation is predominantly lateral in controls and medial in mutants. Dotted line demarks representative area quantified to measure Ppk23$^{glut}$-specific response (see Materials and methods). (D) Low salt attraction of *IR76b* mutants, heterozygous controls, and cell-specific rescue of *IR76b* in Gr64f sweet neurons. Preference measured in binary choice assay with 50 mM NaCl plus 2 mM sucrose versus 2 mM sucrose alone. Bars represent mean ±SEM. n = 30 – 40 groups of 10 flies each, with filled grey circles (controls), open red circles (*IR76b* mutants), or filled red circles (rescues) indicating values for individual groups. Asterisks denote significance by one-way ANOVA with Bonferroni post hoc test, $***p<0.001$. (E) High salt avoidance of *IR76b* mutants, heterozygous controls, and cell-specific rescue of IR76b in Gr66a bitter neurons or Ppk23 neurons. Preference measured in binary choice assay with 250 mM NaCl plus 25 mM sucrose versus 5 mM sucrose alone. Bars represent mean ±SEM. n = 22 – 52 groups of 10 flies each, with filled grey circles (controls), open red circles (*IR76b* mutants), or filled red circles (rescues) indicating values for individual groups. Asterisks denote significance by one-way ANOVA with Bonferroni post hoc test, $***p<0.001$.

DOI: https://doi.org/10.7554/eLife.37167.012

The following source data and figure supplements are available for figure 4:

**Source data 1.** Raw numerical data for *Figure 4* and associated figure supplements.
DOI: https://doi.org/10.7554/eLife.37167.016

**Figure supplement 1.** Ppk23 GRN salt responses require *IR76b*.
DOI: https://doi.org/10.7554/eLife.37167.013

**Figure supplement 2.** A subset of GRN salt responses requires *IR25a*.
DOI: https://doi.org/10.7554/eLife.37167.014

**Figure supplement 3.** Ppk23 GRN salt responses do not require *Ppk23* or *Ppk29*.
DOI: https://doi.org/10.7554/eLife.37167.015

## The cellular basis for salt attraction and avoidance

To probe the cellular basis of salt behaviors, we conditionally silenced different GRN populations using Kir2.1 expression temporally restricted with Gal80$^{ts}$. As expected, both Ppk23$^{glut}$ and Ppk28 GRNs were dispensable for low salt attraction (*Figure 5A*). Focusing on the two GRN classes with low salt tuning properties, we found that Gr64f GRN activity is necessary for attraction to 50 mM NaCl, but expression of Kir2.1 in IR94e GRNs had no effect (*Figure 5A*). Further, silencing Gr64f and IR94e neurons together resulted in behavior indistinguishable from Gr64f silencing alone. Puzzled by the apparent lack of a role for IR94e GRNs in salt attraction, we expressed a different effector – tetanus toxin (TNT) – in these neurons without any temporal restriction with Gal80$^{ts}$, and observed reduced low salt attraction (*Figure 5—figure supplement 1*). Moreover, attraction was virtually eliminated when TNT was expressed in both Gr64f and IR94e GRNs (*Figure 5—figure supplement 1*). Thus, we conclude that sweet GRNs likely mediate the bulk of low salt attraction, with additional input from the IR94e class.

Interestingly, Kir2.1 expression in *IR76b-Gal4* GRNs had a similar effect to Gr64f silencing, suggesting that Gr64f mediates the bulk of IR76b-dependent low salt attraction. However, this phenotype appears less severe than that of *IR76b* mutants, which display mild low salt avoidance (*Figure 4D*). This could reflect incomplete silencing from Kir2.1, as suggested by the lack of observable effects in IR94e GRNs, or weak IR76b-independent low salt responses in Gr66a (bitter/high salt) GRNs that further reduce salt preference in *IR76b* mutants. In any case, restoring *IR76b* selectively to Gr64 neurons rescues low salt attraction in *IR76b* mutants, further supporting the role of sweet neurons in salt attraction (*Figure 4D*).

Since Gr64f neurons are necessary for sugar detection, we sought verification that the Gr64f salt attraction phenotype was not from an inability to sense the low concentration of sucrose in both food options. Indeed, Gr64f silencing caused a similar reduction in salt attraction in the absence of sugar (*Figure 5B*).

We then tested the role of each high salt GRN class in high salt avoidance and found that Gr66a and Ppk23$^{glut}$ GRNs are both necessary for this behavior (*Figure 5C*). To confirm the novel role for Ppk23$^{glut}$ in behavioral salt avoidance, we tested its impact on the Proboscis Extension Reflex (PER), which is an acute measure of gustatory palatability. Consistent with our binary choice assay, silencing

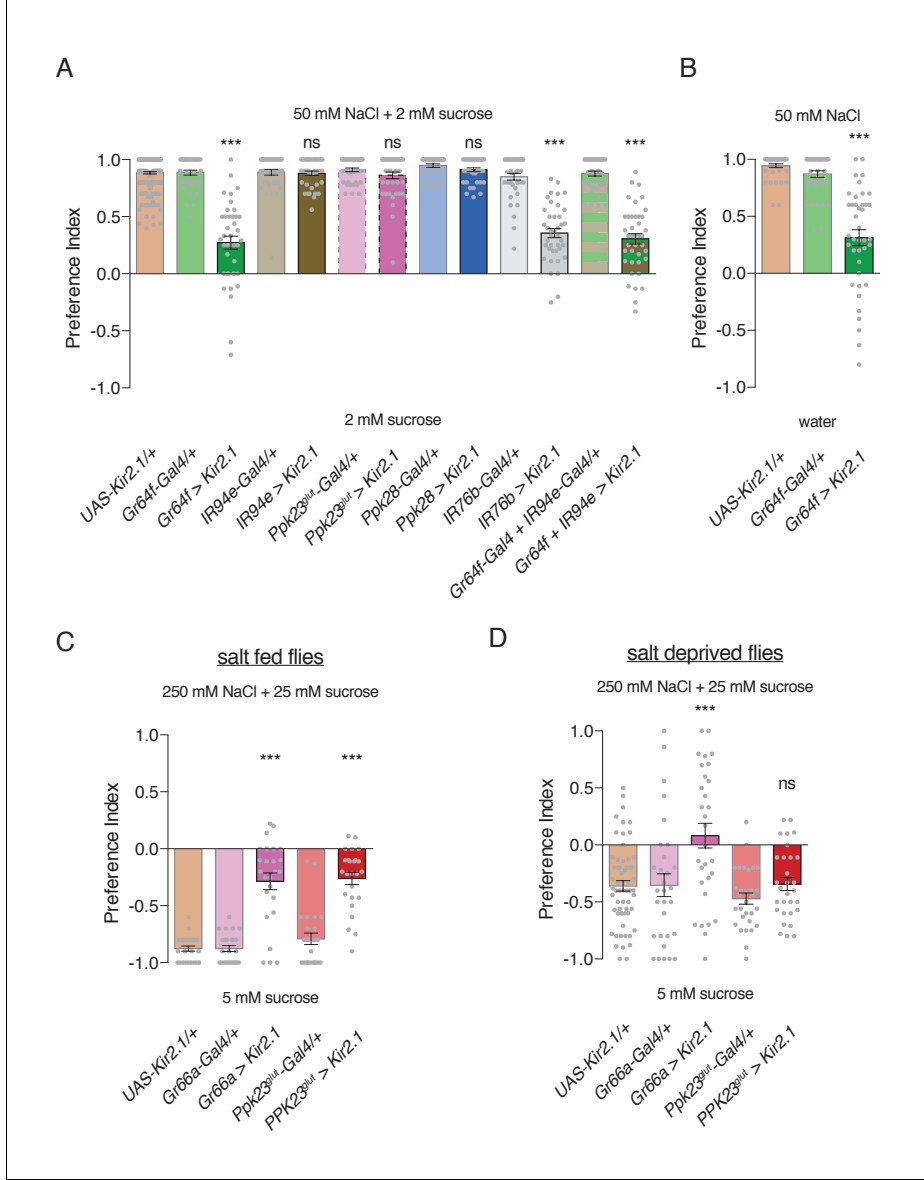

**Figure 5.** Specific GRN contributions to salt attraction and avoidance. (**A**) Low salt attraction in binary choice assay, following silencing of different GRN populations with Kir2.1. Positive values indicate preference for 50 mM NaCl plus 2 mM sucrose; negative values indicate preference for 2 mM sucrose alone. Bars represent mean ±SEM. n = 40 groups of 10 flies each for all genotypes except the *UAS-Kir2.1/+* control, where n = 200. Filled grey circles indicate values for individual groups. Asterisks denote significant difference from both *UAS-Kir2.1/+* and corresponding *Gal4/+* controls by one-way ANOVA with Bonferroni post hoc test, ***p<0.001. (**B**) Low salt attraction tested in the absence of sugar. Bars represent mean ±SEM. n = 40 groups of 10 flies each. Asterisks denote significant difference from both *UAS-Kir2.1/+* and corresponding *Gal4/+* controls by one-way ANOVA with Bonferroni post hoc test, ***p<0.001. (**C–D**) Requirements of aversive GRNs in high salt avoidance under salt fed (**C**) or salt deprived (**D**) conditions. Positive values indicate preference for 250 mM NaCl plus 25 mM sucrose; negative values indicate preference for 5 mM sucrose. Bars represent mean ±SEM. n = 25 (**C**) or 30 (**D**) groups of 10 flies each for all genotypes except the *UAS-Kir2.1/+* control in (**D**), where n = 60. Filled grey circles indicate values for individual groups. Asterisks denote significant difference from both *UAS-Kir2.1/+* and corresponding *Gal4/+* controls by one-way ANOVA with Bonferroni post hoc test, ***p<0.001. For all panels, 'Ppk23$^{glut}$-Gal4' indicates *Ppk23-Gal4* with *Gr66a-LexA* and *LexAop-Gal80*; and '>' denotes indicated Gal4 driving *UAS-Kir2.1* with temporal restriction by *tub-Gal80$^{ts}$*.

DOI: https://doi.org/10.7554/eLife.37167.017

The following source data and figure supplements are available for figure 5:

*Figure 5 continued on next page*

*Figure 5 continued*

**Source data 1.** Raw numerical data for *Figure 5* and associated figure supplements.
DOI: https://doi.org/10.7554/eLife.37167.021
**Figure supplement 1.** Silencing with tetanus toxin reveals a role for IR94e in low salt attraction.
DOI: https://doi.org/10.7554/eLife.37167.018
**Figure supplement 2.** Ppk23$^{glut}$ and Gr66a GRNs differentially function in PER suppression by high salt, depending on internal state.
DOI: https://doi.org/10.7554/eLife.37167.019
**Figure supplement 3.** Ppk23$^{glut}$ calcium responses are not modulated by salt deprivation.
DOI: https://doi.org/10.7554/eLife.37167.020

Ppk23$^{glut}$ GRNs severely impaired the inhibition of PER by high salt (*Figure 5—figure supplement 2A*). Moreover, rescue of *IR76b* expression in either Gr66a or Ppk23$^{glut}$ GRNs partially restores high salt avoidance to *IR76b* mutants (*Figure 4E*).

## Ppk23$^{glut}$ mediates state-dependent modulation of salt behaviors

Fly gustatory responses are frequently modulated by need for specific nutrients (*Kim et al., 2017*). However, modulating salt behaviors presents a complex problem because two of the three GRN classes exhibiting strong salt-evoked activity – Gr64f (sweet) and Gr66a (bitter) – have prominent roles in the detection of other modalities. These are therefore poor candidates for need-dependent modulation of salt responses, unless plasticity is achieved by regulating a salt-specific receptor. We therefore speculated that Ppk23$^{glut}$ GRNs, which to our knowledge specifically respond to salt, may tune the fly's salt behaviors based on need.

The high salt assay shown in *Figure 5C* was performed on flies under salt fed conditions (three days with food containing 10 mM NaCl) to maximize salt avoidance. We subsequently repeated this experiment with flies deprived of salt for three days and observed the expected weakening of salt aversion in controls (*Figure 5D*; p<0.0001 compared to *Figure 5C*). Strikingly, while silencing Gr66a GRNs further reduced salt avoidance, silencing Ppk23$^{glut}$ GRNs had no effect (*Figure 5D*). This suggests that the aversiveness of Ppk23$^{glut}$ GRN activation is suppressed by salt deprivation. To verify this result, we again turned to PER and found that salt deprivation reduced high salt inhibition of PER and suppressed the role of Ppk23$^{glut}$ GRNs (*Figure 5—figure supplement 2A*). Interestingly, Gr66a GRN silencing produced only weak effects on PER inhibition by high salt, which were significantly manifested only in the salt deprived state (*Figure 5—figure supplement 2B*).

We next asked whether the observed behavioral modulation by salt deprivation would be evident in the calcium responses of these neurons; however, salt deprivation led to only a very mild and statistically insignificant reduction in Ppk23$^{glut}$ salt responses (*Figure 5—figure supplement 3*). Therefore, modulation is likely to occur downstream of GRN output.

To further explore this idea, we built a closed-loop system for real-time optogenetic activation of neurons during feeding behavior. Developed as an add-on to the fly Proboscis and Activity Detector (FlyPAD; [*Itskov et al., 2014*]), our system triggers illumination of a red LED immediately upon detecting a fly's interaction with one of the two food sources (*Figure 6A*). We call this system the Sip-TRiggered Optogenetic Behavior Enclosure (STROBE). The STROBE is similar in concept to another recently described optogenetic FlyPAD (*Steck et al., 2018*), but implements sip detection and light triggering in a different way to minimize latency and achieve illumination during sips, with LED activation tightly locked to sip onset and offset.

As expected, sip-induced triggering of Gr64f GRN activation makes a tasteless food source attractive compared to the same food without light stimulation, and this effect is independent of salt deprivation (*Figure 6B*). Similarly, Gr66a activation is strongly aversive for both salt fed and salt deprived flies. Consistent with their lack of a strong phenotype when silenced, activation of IR94e neurons did not produce a detectable phenotype in either condition. However, stimulating Ppk23$^{glut}$ GRNs is aversive, but only when flies have been pre-fed on a salt-containing diet (*Figure 6B*). This supports a model where salt need modulates salt avoidance downstream of Ppk23$^{glut}$ GRN activation.

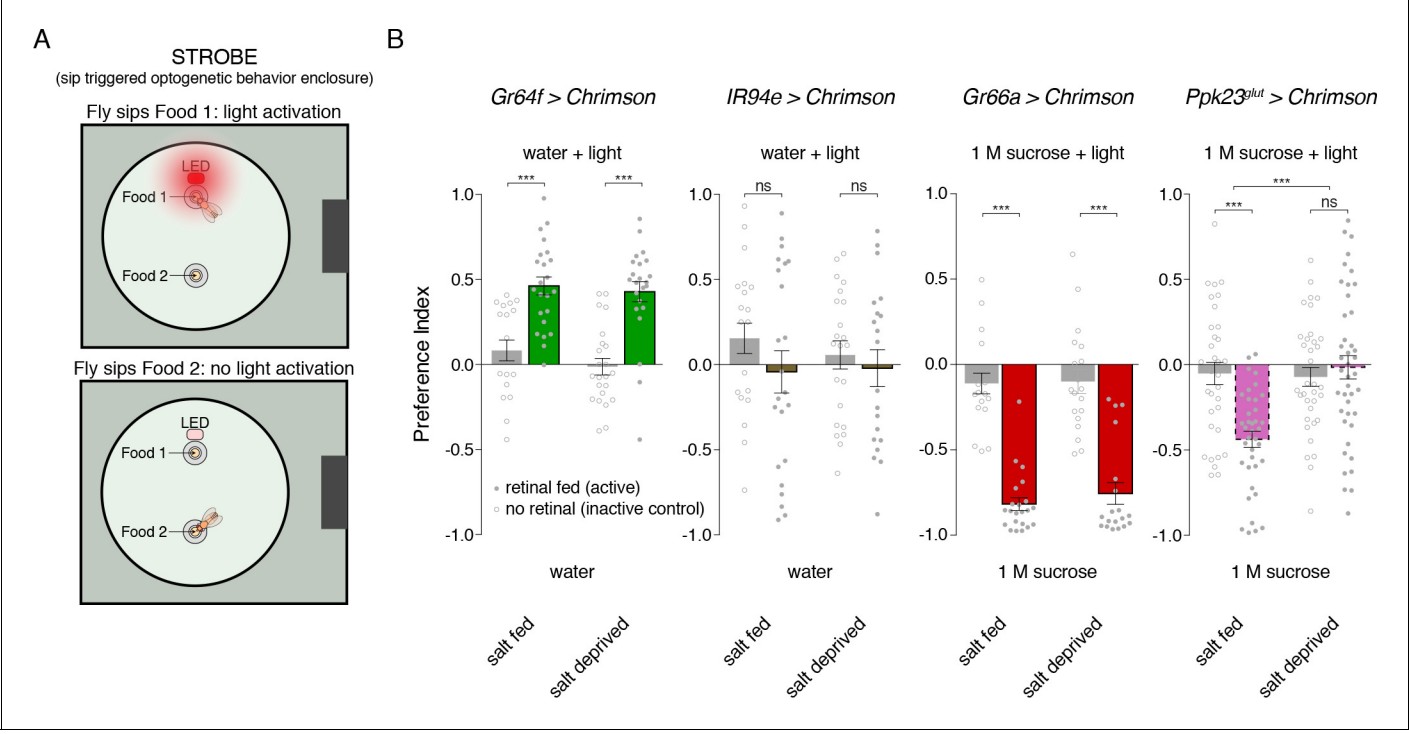

**Figure 6.** Salt deprivation modulates salt avoidance downstream of Ppk23glut. (**A**) Schematic of closed-loop optogentic feeding assay, called the 'sip triggered optogenetic behavior enclosure' (STROBE). Each food is presented as a small drop containing 1% agar. Interactions with Food 1 that are recorded as 'sips' trigger illumination of an LED in the roof of the enclosure, above Food 1. Light triggering is temporally coupled to sip onset with minimal latency. Sips on Food 2 are recorded, but no light is triggered. (**B**) STROBE results for optogenetic stimulation of each salt-responsive GRN class. CsChrimson was expressed in each indicated GRN class, and the preference for Food 1 (light triggering) vs Food 2 (no light triggered) is plotted. Food 1 and Food 2 were otherwise the same in each experiment: plain agar ('water') for Gr64f and IR94e activation, and 1M sucrose for Gr66a and Ppk23glut activation. 1 M sucrose was used for aversive GRN tests because we observe more robust avoidance in this context. Bars represent mean ±SEM. Colored bars represent flies fed retinal (active CsChrimson) and gray bars represent flies not fed retinal (inactive controls). n = 19 – 24 for Gr64f, IR94e and Gr66a experiments, and n = 34 – 42 for Ppk23glut experiment. Filled (retinal fed) or open (not retinal fed) circles indicate values for individual flies. Asterisks indicate significant differences between retinal and no retinal groups for each condition by two-way ANOVA with Bonferroni post hoc test, ***p<0.001. Asterisks between fed and deprived conditions for Ppk23glut experiment represent a significant interaction between salt feeding and ±retinal conditions, ***p<0.001.

DOI: https://doi.org/10.7554/eLife.37167.022

The following source data is available for figure 6:

**Source data 1.** Raw numerical data for *Figure 6*.
DOI: https://doi.org/10.7554/eLife.37167.023

## Discussion

Our results suggest a complex model for how the fly peripheral gustatory system encodes salt taste. In contrast to other known taste modalities, which typically activate a single, molecularly defined population of sensory neurons, salt taste is encoded by the combined activity of most to all GRN classes (*Figure 7*). By identifying markers that cumulatively cover virtually every labellar GRN, we find that two cell types – Gr64f and IR94e – display low salt tuning properties and mediate salt attraction. Although Gr64f neurons are also activated by higher concentrations of salt, their relatively low threshold for activation and specificity for sodium are consistent with a low salt GRN identity. Two other cell types – Gr66a and Ppk23glut – act as high salt GRNs, responding ion non-selectively to high concentrations of salt and driving avoidance. Moreover, the impact of Ppk23glut activation is suppressed upon salt deprivation, providing a means to reduce salt avoidance when need is elevated.

Prior salt coding models have primarily relied on correlations between neural activity and behavioural responses to different salt stimuli, as well as changes to those properties in mutants that may

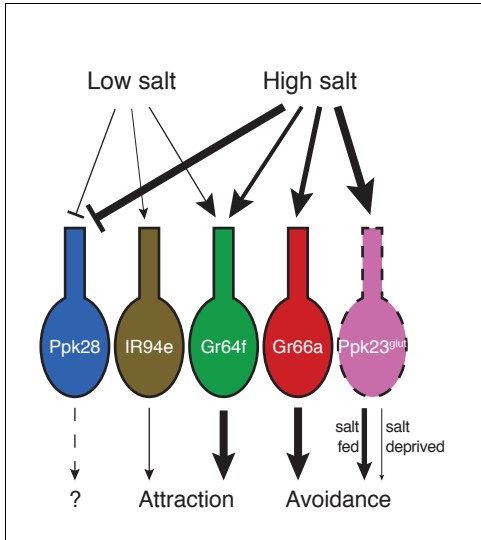

**Figure 7.** Model for salt encoding across different GRN classes in the labellum. Line thickness indicates strength of the excitatory (arrows) or inhibitory (bars) effects of high and low salt on each GRN class, as well as the impact of each cell type on behavior.
DOI: https://doi.org/10.7554/eLife.37167.024

have wide ranging effects (*Ishimoto and Tanimura, 2004*; *Lee et al., 2017*; *Zhang et al., 2013*). These studies led to the important idea that there are distinct low salt and high salt cells present, and that at increasing salt concentrations, the aversive high salt cell progressively dominates over the attractive low salt cell. However, without the molecular tools to identify and manipulate individual GRN classes, the identity and number of salt-responsive cell types were unclear. Our examination of salt coding across a comprehensive set of molecularly defined GRN classes provides new insight into the complexity of salt coding in flies, but is not without limitations. Most notably, we cannot detect possible heterogeneity within defined GRN classes. For example, a Gr64f (sweet) neuron is present in each bristle of all sensillum types. Because we looked at the population as a whole, we cannot confidently conclude that every Gr64f cell acts as a low salt cell; we know only that there are sodium-specific salt responses from the Gr64f population, and that this population drives low salt attraction.

## Ppk23 labels a new high salt cell

Electrophysiological recordings of individual labellar taste sensilla identified high salt responses in the bitter-sensing neurons of S- and I-type sensilla, and previous GRN calcium imaging confirmed that Gr66a neurons respond to 1 M NaCl and KCl (*Marella et al., 2006*; *Meunier et al., 2003*). However, two key results suggested that bitter GRNs did not account for all high salt taste. First, high salt neurons have been identified in L-type sensilla (which don't have Gr66a neurons) via tip recordings, although this has subsequently been debated (*Hiroi et al., 2002*; *Ishimoto and Tanimura, 2004*; *Zhang et al., 2013*). Second, genetically ablating Gr66a GRNs did not block the inhibition of PER by high salt (*Wang et al., 2004*). The existence of Ppk23[glut] high salt cells likely explains both of these observations and provides a mechanism by which flies can specifically modulate their salt behavior in response to need.

In addition to the modulation of their behavioral impact, Ppk23[glut] neurons display some notable characteristics, the most conspicuous being they are the only GRN class to express a marker for glutamatergic, rather than cholinergic, neurons. This adds a potential new dimension to the gustotopic GRN map formed in the fly brain and may be a key mechanism by which the output of Ppk23[glut] neurons remains functionally distinct from other GRN classes targeting postsynaptic neurons in the same area. Indeed, the aversive nature of Ppk23[glut] output stands in contrast to what one would predict from their projection morphology, which looks qualitatively similar to known appetitive (Gr64f, Ppk28), rather than aversive (Gr66a) GRNs. It is possible that Ppk23[glut] aversiveness is mediated through inhibition of appetitive taste pathways, as glutamate can have excitatory or inhibitory postsynaptic effects, depending on the receptor present (*Liu and Wilson, 2013*).

Although Ppk23[glut] defines a novel high salt cell, it is important to note that the Ppk23 channel is not required for its salt responsiveness. This raises questions about what Ppk23-dependent responses these cells may exhibit. Since Ppk23 is required for leg GRN pheromone-evoked activity that regulates courtship, a related function for Ppk23 labellar GRNs cannot be excluded. Indeed, weak, but significant Ppk23-dependent pheromone responses have been observed in labellar GRNs, although it's unclear whether these were from Ppk23[glut] or Ppk23[chat] cells (*Thistle et al., 2012*). Moreover, the interaction between salt taste and mating suggests that perhaps there is a need to co-modulate salt and social cues based on salt diet (*Walker et al., 2015*).

In contrast to the strong salt responses in Ppk23[glut] cells, the other uncharacterized GRN class we identified, IR94e, displayed only weak salt-evoked activity. We therefore expect that this class

primarily responds to other, yet unidentified, taste ligands. Given the lack of strong effects we observe upon activation of IR94e GRNs in the STROBE, we also suspect that the behavioral impact of IR94e activation is, like Ppk23$^{glut}$, state- or context-dependent.

## IR76b has widespread roles in chemosensation

To date, IR76b has been shown to be necessary for gustatory responses to low salt, high salt, calcium, acids, amino acids, fatty acids, and polyamines (*Ahn et al., 2017b*; *Chen and Amrein, 2017*; *Hussain et al., 2016b*; *Lee et al., 2017*; *Zhang et al., 2013*). Consistent with these widespread roles in the taste system, we find expression of *IR76b-Gal4* in every GRN type tested. Nonetheless, we felt it important to clarify the role of IR76b in salt taste, given the apparent complexities in salt responses across labellar GRN types, and the previous demonstration that IR76b can function as a sodium leak channel (*Zhang et al., 2013*).

We find that Gr64f salt responses are completely dependent on IR76b, consistent with its proposed role in low salt taste. Ppk23$^{glut}$ salt responses also require IR76b, but those in Gr66a GRNs do not, indicating two different salt transduction mechanisms in these two high salt cells. This may explain why prior reports differed on whether high salt responses remain intact in *IR76b* mutants (*Lee et al., 2017*; *Zhang et al., 2013*). Interestingly, the IR76b-dependent salt responses in Ppk23$^{glut}$ GRNs are not sodium specific, as we see loss of high sodium and potassium salt-evoked activity. This suggests that, although IR76b is primarily permeable to sodium when expressed in heterologous cells ($P_{Na}$: $P_K$ = 1: 0.4), it may function in complexes with other subunits that confer different ion selectivity in different GRN classes (*Zhang et al., 2013*).

Recently, Ppk23 GRNs were identified as underlying IR76b-dependent calcium taste avoidance (*Lee et al., 2018*). Although it isn't clear whether Ppk23$^{glut}$ or Ppk23$^{chat}$ (or both) subpopulations are responsible, our results indicate that this effect is not specific to calcium, but rather a general salt avoidance mechanism. Indeed, Ppk23 GRNs respond to high concentrations of all salts tested. Moreover, we find that IR25a, which was implicated in Ppk23-mediated calcium taste (*Lee et al., 2018*), is necessary for salt responses in Gr64f and Ppk23$^{glut}$ GRNs, similar to the requirements for IR76b. This stands in contrast to results reported by *Zhang et al. (2013)*, which suggested that IR25a did not play a role in sodium taste. The similar requirements for IR76b and IR25a also suggest that these two receptors may act in a complex to mediate salt taste, which is consistent with previous evidence that IR25a is a broadly expressed coreceptor (*Ahn et al., 2017a*; *Benton et al., 2009*; *Cameron et al., 2010*; *Chen and Amrein, 2017*; *Lee et al., 2018*).

## Modulation of salt avoidance by internal state

Changes in gustatory sensitivity based on internal state are a widespread feature of the fly taste system: starvation potentiates sweet GRN sensitivity and suppresses bitter GRN responses; mating increases taste peg GRN sensitivity to polyamines and behavioral sensitivity to low salt in females; and protein deprivation sensitizes taste peg GRNs to yeast and increases behavioral sensitivity to amino acids (*Hussain et al., 2016a*; *Inagaki et al., 2012*; *Inagaki et al., 2014*; *LeDue et al., 2016*; *Steck et al., 2018*; *Toshima and Tanimura, 2012*; *Walker et al., 2015*). Although modulation of salt taste has not been previously examined in flies, salt depletion in humans increases salt palatability (*Beauchamp et al., 1990*). In line with all these results, we observe significant modulation of fly salt taste behavior by salt deprivation.

In contrast to most taste modalities, which activate a single GRN population, modulation of salt taste presents a complicated problem, because tuning the gain of Gr64 or Gr66a GRN output would have side effects on sweet and bitter taste sensitivity that may be situationally inappropriate. Here, we have presented evidence that the fly gustatory system solves this problem by specifically modulating the effects downstream of Ppk23$^{glut}$ activation. Salt deprivation suppresses the aversiveness of these neurons, allowing the fly to be less repulsed (or more attracted) to salty foods.

Thus, the fly taste system appears to encode salt as a complex mixture of attractive and repulsive sensory responses. Two GRN classes – Gr64f and Gr66a – provide a baseline level of attraction or avoidance, and this response is then adjusted to need via modulation of a third class of salt-responsive GRNs, Ppk23$^{glut}$. The apparent specificity of labellar Ppk23$^{glut}$ GRNs to salt may also provide an important neural substrate for discrimination between salt and other taste modalities. Continued exploration of how salt, and other, taste signals are integrated higher in the brain will provide insight

into how an apparently low-dimensional sensory system can successfully encode a variety of diverse chemical cues.

# Materials and methods

## Key resources table

| Reagent type (species) or resource | Designation | Source or reference | Identifiers | Additional information |
|---|---|---|---|---|
| Genetic reagent (D. melanogaster) | vGlut$^{MI04979}$-Gal4 | Diao et al. (2015) | BDSC: 60312; RRID:BDSC_60312 | |
| Genetic reagent (D. melanogaster) | ChAT$^{MI04508}$-Gal4 | Diao et al. (2015) | BDSC: 60317; RRID:BDSC_60317 | |
| Genetic reagent (D. melanogaster) | vGlut$^{MI04979}$-LexA::QFAD | Diao et al. (2015) | BDSC: 60314; RRID:BDSC_60314 | |
| Genetic reagent (D. melanogaster) | ChAT$^{MI04508}$-LexA::QFAD | Diao et al. (2015) | BDSC: 60319; RRID:BDSC_60319 | |
| Genetic reagent (D. melanogaster) | vGlut$^{MI04979}$-Gal80 | Diao et al. (2015) | BDSC: 60316; RRID:BDSC_60316 | |
| Genetic reagent (D. melanogaster) | ChAT$^{MI04508}$-Gal80 | Diao et al. (2015) | BDSC: 60321; RRID:BDSC_60321 | |
| Genetic reagent (D. melanogaster) | ΔPpk23 | Thistle et al. (2012) | Flybase: FBal0277047 | |
| Genetic reagent (D. melanogaster) | ΔPpk29 | Thistle et al. (2012) | Flybase: FBal0277049 | |
| Genetic reagent (D. melanogaster) | Gr66a-LexA | Thistle et al. (2012) | Flybase: FBal0277069 | |
| Genetic reagent (D. melanogaster) | ppk28-LexA | Thistle et al. (2012) | Flybase: FBal0277050 | |
| Genetic reagent (D. melanogaster) | ppk23-Gal4 | Thistle et al. (2012) | Flybase: FBal0277044 | |
| Genetic reagent (D. melanogaster) | Gr64f$^{LexA}$ | Miyamoto et al. (2012) | Flybase: FBti0168176 | |
| Genetic reagent (D. melanogaster) | ppk23-LexA | Toda et al. (2012) | Flybase: FBst0051311 | |
| Genetic reagent (D. melanogaster) | IR76b-Gal4 | Zhang et al. (2013) | Flybase: FBtp0085485 | |
| Genetic reagent (D. melanogaster) | IR76b$^1$ | Zhang et al. (2013) | Flybase: FBst0051309 | |
| Genetic reagent (D. melanogaster) | IR76b$^2$ | Zhang et al. (2013) | Flybase: FBst0051310 | |
| Genetic reagent (D. melanogaster) | UAS-IR76b | Zhang et al. (2013) | Flybase: FBtp0085485 | |
| Genetic reagent (D. melanogaster) | IR25a$^1$ | Benton et al. (2009) | Flybase: FBst0041736 | |
| Genetic reagent (D. melanogaster) | IR25a$^2$ | Benton et al. (2009) | Flybase: FBst0041737 | |
| Genetic reagent (D. melanogaster) | UAS-IR25a | Abuin et al., 2011 | Flybase: FBst0041747 | |
| Genetic reagent (D. melanogaster) | Gr66a-Gal4 | Wang et al. (2004) | Flybase: FBtp0014660 | |
| Genetic reagent (D. melanogaster) | Gr64f-Gal4 | Dahanukar et al. (2007) | Flybase: FBti0162678 | |
| Genetic reagent (D. melanogaster) | Gr64f-Gal4 | Dahanukar et al. (2007) | Flybase: FBtp0057275 | |
| Genetic reagent (D. melanogaster) | Ppk28-Gal4 | Cameron et al. (2010) | Flybase: FBtp0054514 | |

*Continued on next page*

*Continued*

| Reagent type (species) or resource | Designation | Source or reference | Identifiers | Additional information |
|---|---|---|---|---|
| Genetic reagent (*D. melanogaster*) | *LexAop-CD2::GFP* | *Lai and Lee (2006)* | Flybase: FBti0186090 | |
| Genetic reagent (*D. melanogaster*) | *UAS-Kir2.1* | *Baines et al. (2001)* | Flybase: FBti0017552 | |
| Genetic reagent (*D. melanogaster*) | *tub-Gal80*[ts] | *McGuire et al. (2004)* | Flybase: FBti0027797 | |
| Genetic reagent (*D. melanogaster*) | *IR94e-Gal4* | *Tirián and Dickson, 2017* | VDRC: v207582 | |
| Genetic reagent (*D. melanogaster*) | *w*[1118] | Bloomington Drosophila Stock Center | BDSC: 3605; RRID:BDSC_3605 | |
| Genetic reagent (*D. melanogaster*) | *LexAop-Gal80* | Bloomington Drosophila Stock Center | BDSC: 32214; RRID:BDSC_32214 | |
| Genetic reagent (*D. melanogaster*) | *LexAop-GCaMP6f* | Bloomington Drosophila Stock Center | BDSC: 44277; RRID:BDSC_44277 | |
| Genetic reagent (*D. melanogaster*) | *UAS-GCaMP6f* | Bloomington Drosophila Stock Center | BDSC: 42747; RRID:BDSC_42747 | |
| Genetic reagent (*D. melanogaster*) | *UAS-GCaMP6f* | Bloomington Drosophila Stock Center | BDSC: 52869; RRID:BDSC_52869 | |
| Genetic reagent (*D. melanogaster*) | *UAS-CsChrimson* | Bloomington Drosophila Stock Center | BDSC: 55135; RRID:BDSC_55135 | |
| Genetic reagent (*D. melanogaster*) | *UAS-TNT* | Bloomington Drosophila Stock Center | BDSC: 28838; RRID:BDSC_28838 | |
| Genetic reagent (*D. melanogaster*) | *UAS-impTNT* | Bloomington Drosophila Stock Center | BDSC: 28840; RRID:BDSC_28840 | |
| Antibody | anti-GFP | Abcam, Cambridge, UK, | #13970; RRID:AB_300798 | (1:1000 dilution) |
| Antibody | anti-RFP | Rockland Immunochemicals, Pottstown, PA, | #600-401-379; RRID:AB_2209751 | (1:200 dilution) |
| Antibody | anti-chicken Alexa 488 | Abcam | #150169; RRID:AB_2636803 | (1:200 dilution) |
| Antibody | anti-rabbit Alexa 647 | Thermo Fisher Scientific, Waltham, MA, | #A21245; RRID:AB_2535813 | (1:200 dilution) |
| Antibody | anti-brp | Developmental Studies Hybridoma Bank | #nc82; RRID:AB_2314866 | (1:50 dilution) |
| Antibody | anti-rabbit Alexa 568 | Thermo Fisher Scientific, Waltham, MA, | #A11036; RRID:AB_10563566 | (1:200 dilution) |
| Chemical compound, drug | All trans-Retinal | Sigma-Aldrich | #R2500 | |
| Chemical compound, drug | Sucrose | Sigma-Aldrich | #S7903 | |
| Chemical compound, drug | NaCl | Sigma-Aldrich | #S7653 | |
| Chemical compound, drug | KCl | Sigma-Aldrich | #P9541 | |

*Continued on next page*

*Continued*

| Reagent type (species) or resource | Designation | Source or reference | Identifiers | Additional information |
|---|---|---|---|---|
| Chemical compound, drug | NaBr | Sigma-Aldrich | #S4547 | |
| Chemical compound, drug | KBr | Sigma-Aldrich | #221864 | |
| Chemical compound, drug | CsCl | Sigma-Aldrich | #289329 | |
| Chemical compound, drug | CaCl2 | BDH chemicals | #BDH4524 | |
| Chemical compound, drug | Lobeline hydrochloride | Sigma-Aldrich | #141879 | |
| Chemical compound, drug | Caffeine | Sigma-Aldrich | #C0750 | |
| Chemical compound, drug | 7,11-heptacosadiene (7,11-HC) | Caymen chemical company | #10012567 | |
| Chemical compound, drug | 7,11-nonacosadiene (7,11-NC) | Caymen chemical company | #9000314 | |
| Chemical compound, drug | 7-tricosene (7 T) | Caymen chemical company | #9000313 | |
| Chemical compound, drug | Cis-vaccenyl acetate (c-VA) | Caymen chemical company | #10010101 | |
| Chemical compound, drug | Erioglaucine | Spectrum chemical | #FD110 | |
| Chemical compound, drug | Amaranth | Sigma-Aldrich | #A1016 | |
| Software, algorithm | STROBE executable | *Chan, 2018a* | github: https://github.com/rcwchan/STROBE_software/ (copy archived at https://github.com/elifesciences-publications/STROBE_software) | |
| Software, algorithm | STROBE post-processing | *Chan, 2018a* | github: https://github.com/rcwchan/STROBE_software/ (copy archived at https://github.com/elifesciences-publications/STROBE_software) | |
| Software, algorithm | STROBE VHDL code | *Chan, 2018b* | github: https://github.com/rcwchan/STROBE-fpga (copy archived at https://github.com/elifesciences-publications/STROBE-fpga) | |
| Software, algorithm | ImageJ | *Schneider et al. (2012)* | https://imagej.nih.gov/ij; RRID:SCR_003070 | |
| Software, algorithm | Prism 6 | Graphpad | RRID:SCR_002798 | |
| Software, algorithm | Photoshop | Adobe | RRID:SCR_014199 | |
| Software, algorithm | Illustrator | Adobe | RRID:SCR_010279 | |

## Fly genotype table

| Figure panel | Genotype |
|---|---|
| *Figure 1C* | +/+; vGlut$^{MI04979}$-Gal4/+; UAS-CD8::tdTomato/+ |
| *Figure 1D* | +/+; vGlut$^{MI04979}$-Gal4/LexAop-CD2::GFP ; UAS-CD8::tdTomato/ChAT$^{MI04508}$- LexA::QFAD |

*Continued on next page*

*Continued*

| Figure panel | Genotype |
| --- | --- |
| *Figure 1E* | +/+; vGlut^MI04979^-Gal4/ LexAop-CD2::GFP; UAS-CD8::tdTomato/Ppk28-LexA |
| *Figure 1F* | +/+; vGlut^MI04979^-Gal4/ LexAop-CD2::GFP; UAS-CD8::tdTomato/Gr64f^LexA |
| *Figure 1G* | +/+; vGlut^MI04979^-Gal4/ LexAop-CD2::GFP; UAS-CD8::tdTomato/Gr66a-LexA |
| *Figure 1H* | +/+; vGlut^MI04979^-Gal4/ LexAop-CD2::GFP; UAS-CD8::tdTomato/Ppk23-LexA |
| *Figure 1I* | +/+; Gr66a-Gal4/ LexAop-CD2::GFP; UAS-CD8::tdTomato/Ppk23-LexA |
| *Figure 1J* | +/+; vGlut^MI04979^-Gal80 /UAS-GCaMP6f; Ppk23-Gal4/+ |
| *Figure 1K* | Gr66a-LexA/+; LexAop-Gal80/ UAS-GCaMP6f; Ppk23-Gal4/+ |
| *Figure 1L* | +/+; UAS-CD8::tdTomato /LexAop-CD2::GFP; IR94e-Gal4/ppk28-LexA |
| *Figure 1M* | +/+; UAS-CD8::tdTomato /LexAop-CD2::GFP; IR94e-Gal4/Gr64f^LexA |
| *Figure 1N* | +/+; UAS-CD8::tdTomato /LexAop-CD2::GFP; IR94e-Gal4/ppk23-LexA |
| *Figure 1—figure supplement 1A* | +/+; ChAT^MI04508^-Gal80 /UAS-GCaMP6f; Ppk23-Gal4/+ |
| *Figure 1—figure supplement 1B* | +/+; UAS-CD8::tdTomato /LexAop-CD2::GFP; Ppk23-Gal4/Ppk23-LexA |
| *Figure 1—figure supplement 1C* | +/+; IR76b-Gal4/ LexAop-CD2::GFP; UAS-CD8::tdTomato/Ppk28-LexA |
| *Figure 1—figure supplement 1D* | +/+; IR76b-Gal4/ LexAop-CD2::GFP; UAS-CD8::tdTomato/Gr64f^LexA |
| *Figure 1—figure supplement 1E* | +/+; IR76b-Gal4 /LexAop-CD2::GFP; UAS-CD8::tdTomato/Gr66a-LexA |
| *Figure 1—figure supplement 1F* | +/+; IR76b-Gal4/LexAop -CD2::GFP; UAS-CD8::tdTomato/ Ppk23-LexA |
| *Figure 1—figure supplement 1G* | +/+; UAS-CsChrimson/+; IR94e-Gal4/+ |
| *Figure 2C and D* | +/+; LexAop-GCaMP6f/+; Ppk28-LexA/+ |
| | +/+; UAS-GCaMP6f/ Gr64f-Gal4; +/+ |
| | +/+; UAS-GCaMP6f/+; IR94e-Gal4/+ |
| | +/+; UAS-GCaMP6f/ Gr66a-Gal4; +/+ |
| | +/+; UAS-GCaMP6f/+; Ppk23-Gal4/+ |
| *Figure 2—figure supplement 1A–D* | +/+; UAS-GCaMP6f/+; Ppk23-Gal4/+ |
| *Figure 2—figure supplement 2* | +/+; UAS-GCaMP6f/+; IR94e-Gal4/+ |
| *Figure 3A* | +/+; UAS-GCaMP6f/+; Ppk23-Gal4/+ |

*Continued on next page*

*Continued*

| Figure panel | Genotype |
|---|---|
| *Figure 3B* | *+/+; UAS-GCaMP6f/+; Ppk23-Gal4/+* |
| | *Gr66a-LexA/+; LexAop-Gal80/UAS-GCaMP6f, Ppk23-Gal4/+* |
| | *+/+; vGlut$^{MI04979}$-Gal80/UAS-GCaMP6f; Ppk23-Gal4/+* |
| *Figure 3C and D* | *Gr66a-LexA/+; LexAop-Gal80/UAS-GCaMP6f, Ppk23-Gal4/+* |
| | *+/+; vGlut$^{MI04979}$-Gal80/UAS-GCaMP6f; Ppk23-Gal4/+* |
| *Figure 3—figure supplement 1A and B* | *+/+; ChAT$^{MI04508}$-Gal80/UAS-GCaMP6f; Ppk23-Gal4/+* |
| *Figure 4A and B* | *+/+; Gr64f-Gal4/UAS-GCaMP6f; IR76b$^2$/+* |
| | *+/+; Gr64f-Gal4/UAS-GCaMP6f; IR76b$^1$/IR76b$^2$* |
| | *+/+; Gr64f-Gal4, UAS-GCaMP6f/UAS-IR76b; IR76b$^1$/IR76b$^2$* |
| | *+/+; Gr66a-Gal4/UAS-GCaMP6f; IR76b$^2$/+* |
| | *+/+; Gr66a-Gal4/UAS-GCaMP6f; IR76b$^1$/IR76b$^2$* |
| | *+/+; Gr66a-Gal4, UAS-GCaMP6f/UAS-IR76b; IR76b$^1$/IR76b$^2$* |
| | *+/+; Ppk23-LexA/LexAop-GCaMP6f; IR76b$^2$/+* |
| | *+/+; Ppk23-LexA/LexAop-GCaMP6f; IR76b$^1$/IR76b$^2$* |
| | *+/+; UAS-GCaMP6f/UAS-IR76b; IR76b$^1$, Ppk23-Gal4/IR76b$^2$* |
| *Figure 4C* | *+/+; Ppk23-LexA/LexAop-GCaMP6f; IR76b$^2$/+* |
| | *+/+; Ppk23-LexA/LexAop-GCaMP6f; IR76b$^1$/IR76b$^2$* |
| *Figure 4D* | *+/+; +/+; IR76b$^1$/+* |
| | *+/+; +/+; IR76b$^2$/+* |
| | *+/+; +/UAS-IR76b; IR76b$^1$/IR76b$^2$* |
| | *+/+; Gr64f-Gal4/+; IR76b$^1$/IR76b$^2$* |
| | *+/+; Gr64f-Gal4/UAS-IR76b; IR76b$^1$/IR76b$^2$* |
| *Figure 4E* | *+/+; +/+; IR76b$^1$/+* |
| | *+/+; +/+; IR76b$^2$/+* |
| | *+/+; +/UAS-IR76b; IR76b$^1$/IR76b$^2$* |
| | *+/+; Gr66a-Gal4/+; IR76b$^1$/IR76b$^2$* |
| | *+/+; Gr66a-Gal4/UAS-IR76b; IR76b$^1$/IR76b$^2$* |
| | *+/+; +/+; Ppk23-Gal4, IR76b$^1$/IR76b$^2$* |

*Continued on next page*

*Continued*

| Figure panel | Genotype |
|---|---|
| | +/+; +/UAS-IR76b; Ppk23-Gal4, IR76b$^1$/IR76b$^2$ |
| *Figure 4—figure supplement 1A and B* | +/+; Ppk23-LexA/LexAop-GCaMP6f; IR76b$^2$/+ |
| | +/+; Ppk23-LexA/LexAop-GCaMP6f; IR76b$^1$/IR76b$^2$ |
| | +/+; UAS-GCaMP6f/UAS-IR76b; IR76b$^1$, Ppk23-Gal4/IR76b$^2$ |
| *Figure 4—figure supplement 2* | +/+; IR25a$^1$/+; Gr64f-Gal4/UAS-GCaMP6f |
| | +/+; IR25a$^1$/IR25a$^2$; Gr64f-Gal4/UAS-GCaMP6f |
| | +/+; UAS-IR25a, IR25a$^1$/IR25a$^2$; Gr64f-Gal4/UAS-GCaMP6f |
| | +/+; IR25a$^1$/+; Gr66a-Gal4/UAS-GCaMP6f |
| | +/+; IR25a$^1$/IR25a$^2$; Gr66a-Gal4/UAS-GCaMP6f |
| | +/+; UAS-IR25a, IR25a$^1$/IR25a$^2$; Gr66a-Gal4/UAS-GCaMP6f |
| | +/+; IR25a$^1$/+; Ppk23-Gal4/UAS-GCaMP6f |
| | +/+; IR25a$^1$/IR25a$^2$; Ppk23-Gal4/UAS-GCaMP6f |
| | +/+; UAS-IR25a, IR25a$^1$/IR25a$^2$; Ppk23-Gal4/UAS-GCaMP6f |
| *Figure 4—figure supplement 3A and B* | ΔPpk23/ΔPpk23; ΔPpk29/ΔPpk29; Ppk23-Gal4/UAS-GCaMP6f |
| *Figure 5A* | +/+; +/+; UAS-Kir2.1, tub-Gal80$^{ts}$/+ |
| | +/+; Gr64f-Gal4/+; +/+ |
| | +/+; Gr64f-Gal4/+; UAS-Kir2.1, tub-Gal80$^{ts}$/+ |
| | +/+; +/+; IR94e-Gal4/+ |
| | +/+; +/+; IR94e-Gal4/UAS-Kir2.1, tub-Gal80$^{ts}$ |
| | Gr66a-LexA/+; LexAop-Gal80/+; Ppk23-Gal4/+ |
| | Gr66a-LexA/+; LexAop-Gal80/+; Ppk23-Gal4/UAS-Kir2.1, tub-Gal80$^{ts}$ |
| | +/+; +/+; Ppk28-Gal4/+ |
| | +/+; +/+; Ppk28-Gal4/UAS-Kir2.1, tub-Gal80$^{ts}$ |
| | +/+; IR76b-Gal4/+; +/+ |
| | +/+; IR76b-Gal4/+; UAS-Kir2.1, tub-Gal80$^{ts}$/+ |
| | +/+; Gr64f-Gal4/+; IR94e-Gal4/+ |

*Continued on next page*

*Continued*

| Figure panel | Genotype |
|---|---|
| | +/+; Gr64f-Gal4/+; IR94e-Gal4/UAS-Kir2.1, tub-Gal80$^{ts}$ |
| *Figure 5B* | +/+; +/+; UAS-Kir2.1, tub-Gal80$^{ts}$/+ |
| | +/+; Gr64f-Gal4/+; +/+ |
| | +/+; Gr64f-Gal4/+; UAS-Kir2.1, tub-Gal80$^{ts}$/+ |
| *Figure 5C* | +/+; +/+; UAS-Kir2.1, tub-Gal80$^{ts}$/+ |
| | +/+; Gr66a-Gal4/+; +/+ |
| | +/+; Gr66a-Gal4/+; UAS-Kir2.1, tub-Gal80$^{ts}$/+ |
| | Gr66a-LexA/+; LexAop-Gal80/+; Ppk23-Gal4/+ |
| | Gr66a-LexA/+; LexAop-Gal80/+; Ppk23-Gal4/UAS-Kir2.1, tub-Gal80$^{ts}$ |
| *Figure 5D* | +/+; +/+; UAS-Kir2.1, tub-Gal80$^{ts}$/+ |
| | +/+; Gr66a-Gal4/+; +/+ |
| | +/+; Gr66a-Gal4/+; UAS-Kir2.1, tub-Gal80$^{ts}$/+ |
| | Gr66a-LexA/+; LexAop-Gal80/+; Ppk23-Gal4/+ |
| | Gr66a-LexA/+; LexAop-Gal80/+; Ppk23-Gal4/UAS-Kir2.1, tub-Gal80$^{ts}$ |
| *Figure 5—figure supplement 1* | +/+; UAS-impTNT/+; +/+ |
| | +/+; UAS-TNT/+; +/+ |
| | +/+; +/+; IR94e-Gal4/+ |
| | +/+; UAS-impTNT/+; IR94e-Gal4/+ |
| | +/+; UAS-TNT/+; IR94e-Gal4/+ |
| | +/+; Gr64f-Gal4/+; IR94e-Gal4/+ |
| | +/+; UAS-impTNT/Gr64f-Gal4; IR94e-Gal4/+ |
| | +/+; UAS-TNT/Gr64f-Gal4; IR94e-Gal4/+ |
| *Figure 5—figure supplement 2A* | Gr66a-LexA/+; LexAop-Gal80/+; Ppk23-Gal4/UAS-Kir2.1, tub-Gal80$^{ts}$ |
| | Gr66a-LexA/+; LexAop-Gal80/+; Ppk23-Gal4/+ |
| | +/+; +/+; UAS-Kir2.1, tub-Gal80$^{ts}$/+ |
| *Figure 5—figure supplement 2B* | +/+; Gr66a-Gal4/+; UAS-Kir2.1, tub-Gal80$^{ts}$/+ |
| | +/+; Gr66a-Gal4/+; +/+ |
| | +/+; +/+; UAS-Kir2.1, tub-Gal80$^{ts}$/+ |
| *Figure 5—figure supplement 3* | +/+; UAS-GCaMP6f/+; Ppk23-Gal4/+ |
| *Figure 6B* | +/+; UAS-CsChrimson/Gr64f-Gal4; +/+ |
| | +/+; UAS-CsChrimson/+; IR94e-Gal4/+ |
| | +/+; UAS-CsChrimson/Gr66a-Gal4; +/+ |
| | Gr66a-LexA/+; LexAop-Gal80/UAS-CsChrimson; Ppk23-Gal4/+ |

## Flies

Flies were raised on standard cornmeal fly food at 25°C in 70% humidity. The following genotypes were used: *vGlut$^{MI04979}$-Gal4*, *ChAT$^{MI04508}$-Gal4*, *vGlut$^{MI04979}$-LexA::QFAD*, *ChAT$^{MI04508}$- LexA::*

QFAD, vGlut$^{MI04979}$-Gal80, ChAT$^{MI04508}$-Gal80 (**Diao et al., 2015**); Gr66a-LexA, ppk28-LexA, ppk23-Gal4, UAS-CD8::tdTomato (**Thistle et al., 2012**); Gr64f$^{LexA}$ (**Miyamoto et al., 2012**); ppk23-LexA (**Toda et al., 2012**); IR76b-Gal4, IR76b$^{1}$, IR76b$^{2}$, UAS-IR76b (**Zhang et al., 2013**); IR25a$^{1}$, IR25a$^{2}$ (**Benton et al., 2009**); UAS-IR25a (**Abuin et al., 2011**); Gr66a-Gal4 (**Wang et al., 2004**); Gr64f-Gal4 (**Dahanukar et al., 2007**); Ppk28-Gal4 (**Cameron et al., 2010**); LexAop-CD2::GFP (**Lai and Lee, 2006**); UAS-Kir2.1 (**Baines et al., 2001**); tub-Gal80$^{ts}$ (**McGuire et al., 2004**); IR94e-Gal4 (**Tirián and Dickson, 2017**)(Vienna Drosophila Resource Center: v207582); LexAop-Gal80 (32214), LexAop-GCaMP6f (44217), UAS-GCaMP6f (42747 and 52869), UAS-CsChrimson (55135), UAS-TNT (28838), UAS-impTNT (28840) (Bloomington Stock Center).

## Tastants

The following tastants were used: Sucrose, NaCl, KCl, NaBr, KBr, CsCl, CaCl$_2$, Lobeline hydrochloride, Caffeine (Sigma-Aldrich); 7,11-heptacosadiene (7,11-HC), 7,11-nonacosadiene (7,11-NC), 7-tricosene (7 T), and cis-vaccenyl acetate (c-VA) (Cayman Chemical Company, Ann Arbor, MI). Tastants were mostly kept as 1 M stocks and diluted as needed. Lobeline hydrochloride was kept as a 1.25 mM stock. 7,11-heptacosadiene (7,11-HC), 7,11-nonacosadiene (7,11-NC), and 7-tricosene (7 T) were diluted in water to desired 0.0001 mg/ul. Cis-vaccenyl acetate (c-VA) was diluted to stock solution of 0.01 mg/ul in EtOH, and then diluted in water. All hydrocarbons stocks were kept at −20°C, diluted as needed, and stored at 4°C for up to seven days. 1% of each Ethanol and Hexanol were diluted in a mix with water and kept at 4°C as control solution for pheromone imaging.

## Immunohistochemistry

Immunofluorescence on labella was carried out as described (**Jeong et al., 2016**). Labella were dissected and fixed for 25 min in 4% paraformaldehyde in PBS + 0.2% Triton. After washing with PBS + triton (0.2%; PBST), labella were blocked in 5% NGS diluted with PBST for 40 min. The following primary antibodies were applied and incubated at 4°C overnight: chicken anti-GFP (1:1000, Abcam, Cambridge, UK, #13970) and rabbit anti-RFP (1:200, Rockland Immunochemicals, Pottstown, PA, #600-401-379). After washing for 1 hr, the following secondary antibodies were added for 2 hr: goat anti-chicken Alexa 488 (1:200, Abcam #150169) and goat anti-rabbit Alexa 647 (1:200, Thermo Fisher Scientific, Waltham, MA, #A21245). Labella were washed again for 40 min, placed on slides in SlowFade gold (Thermo Fisher Scientific), with small #1 coverslips as spacers.

Brain immunofluorescence was carried out as described previously (**Chu et al., 2014**). Primary antibodies used were chicken anti-GFP (1:1000, Abcam #13970) and mouse anti-brp (1:50, DSHB #nc82). Secondary antibodies used were goat anti-chicken Alexa 488 (1:200, Abcam #150169) and goat anti-rabbit Alexa 568 (1:200, Thermo Fisher Scientific #A11036).

All images were acquired using a Leica SP5 II Confocal microscope with a 25x water immersion objective. Images were processed in ImageJ (**Schneider et al., 2012**) and Adobe Photoshop.

## Labellar expression map annotation

To annotate the expression of different markers in the labellum, each sensillum was analyzed in 4 – 8 labella stained for each combination of markers. Confocal z-stacks were examined to identify how many neurons in each sensillum were positive for the different drivers, and which neurons overlapped with the respective co-labelled population. The most common result for each neuron in each sensillum was reported. Sensilla S0, I0, I9, and I10 were the most difficult to score because of viewing difficulties. At times there were duplications of specific sensilla on a labellum, in which case both sensilla were considered.

## GCaMP imaging

For calcium imaging experiments, female or male flies were aged from 2 to 10 days in groups of both sexes. Females were used for all experiments except where indicated (pheromones). Prior to imaging, flies were briefly anesthetized using CO2, legs amputated for full access to the proboscis, and placed in custom chamber suspended from their cervix. To ensure immobilization, a small drop of nail polish was applied to the back of the neck and the proboscis was pulled to extension and waxed out on both sides. A modified dental waxer was used to apply wax on each side of the chamber rim, making little contact with the feeding structure. Flies were left to recover in a humidified

chamber for 1 hr. The antenna were removed from the fly and a small window of cuticle was removed from the top of the head, exposing the SEZ. Adult Hymolymph Like (AHL) buffer was immediately applied to the preparation (108 mM NaCl, 5 mM KCl, 4 mM NaHCO3, 1 mM NaH2PO4, 5 mM HEPES, 15 mM ribose, pH 7.5). The air sacs, fat, and esophagus were clipped and removed to allow clear visualization on the SEZ. Once ready to image, AHL buffer was added that includes $Mg^{2+}$ and $Ca^{2+}$ (108 mM NaCl, 5 mM KCl, 4 mM NaHCO3, 1 mM NaH2PO4, 5 mM HEPES, 15 mM ribose, 2 mM $Ca^{2+}$, and 8.2 mM $Mg^{2+}$).

GCaMP6f fluorescence was observed using a Leica SP5 II Confocal microscope with a 25x water immersion objective. The relevant area of the SEZ was visualized at a zoom of 4x, a line speed of 8000 Hz, a line accumulation of 2, and resolution of 512 × 512 pixels. The pinhole was opened to 2.98 AU. For each taste stimulation, data was acquired during a baseline of 5 s prior to stimulation, 1 s during tastant application, and 9 s following the stimulation.

Tastant stimulations were done using a pulled capillary pipette that was filed down to match the size of the proboscis and fit over all taste sensilla on both labellar palps. The pipette was filled with 1 – 2 µl of a tastant and positioned close to the proboscis labellum. At 5 s a micromanipulator was used to apply the tastant to the labellum manually. Between taste stimulations of differing solutions, the pipette was washed with water. All NaCl solutions were applied in the order of increasing concentration, finishing with 1M KCl. All other solutions were applied in random order to control for potential inhibitory effects between modalities.

The maximum change in fluorescence (ΔF/F) was calculated using the peak intensity (average of 3 time points) minus the average intensity at baseline (10 time points), divided by the baseline. Quantification of fluorescence changes was performed in ImageJ and graphed in GraphPad Prism6.

For quantification of Ppk23[glut] projections, the caffeine response for each fly was used to create a region of interest starting below the 'bitter ring' and extending across to encompass the lateral projections. This same region of interest was applied to the salt responses of that fly to exclude the Ppk23[chat] population overlapping with Gr66a in this 'bitter ring'.

## Salt deprivation

Flies were placed in one of two conditions for 2 – 3 days: 1% agar, 5% sucrose, and 10 mM NaCl (salt fed); or 1% agar and 5% sucrose (salt deprived).

## Behavioral assays

Binary choice preference tests were similar to those previously described (*LeDue et al., 2015*). Female flies aged 2–5 days were sorted into groups of 10 and placed in conditions of either salt feeding or salt deprivation (see above) and shifted to 29°C for 48 hr to induce expression of Kir2.1 in the cells of interest. For the low salt assay, salt deprived flies were then tested directly. Flies for the high salt assay were subjected to a subsequent 12 hr on medium without sugar (but with the same salt content) to increase sugar attraction. For both assays, flies were then transferred into testing vials containing six 10 µL dots of agar that alternated in color. For most low salt attraction assays, the food choices were: 1% agar with both 2 mM sucrose and 50 mM NaCl (Food 1), and 1% agar with 2 mM sucrose (Food 2). The experiment in *Figure 5B* was done without sucrose. For the high salt avoidance assays, the food choices were: 1% agar with both 25 mM sucrose and 250 mM NaCl (Food 1), and 1% agar with 5 mM sucrose (Food 2). Each choice contained either 0.125 mg/mL blue (Erioglaucine, FD and C Blue#1) or 0.5 mg/mL red (Amaranth, FD and C Red#2) dye, and half the replicates for each experiment were done with the dyes swapped to control for any dye preference. Flies were allowed to feed for 2 hr in the dark at 29°C and then frozen and scored for abdomen color. Preference index (PI) was calculated as ((# of flies labeled with Food 1 color) – (# of flies labeled with Food 2 color))/(total number of flies that fed).

For PER, 2 – 5 day old females were collected and treated exactly as described above for salt fed and deprived conditions. Flies were then mounted inside pipette tips that were cut to size so that only the head was exposed. The tubes were sealed at the end with tape, positioned on a glass slide with double-sided tape. After a 1 – 2 hr recovery, flies were stimulated with water and allowed to drink until satiated. Each fly was then stimulated on the labellum with increasing concentrations of salt (0 mM salt, 250 mM NaCl, 500 mM NaCl, 1 M NaCl, and 1 M KCl) mixed with 100 mM sucrose using a 20 µL pipette attached to a 1 mL syringe. Stimuli were presented three times each per fly.

Four groups of 10 flies for each genotype were tested over four days. The order of genotypes tested on each day was randomized.

## STROBE system

The STROBE builds on the FlyPAD system's hardware (*Itskov et al., 2014*) by adding a lighting circuit and opaque curtain (to prevent interference from outside lighting) to each of 16 FlyPAD arenas. Thus, together, a functioning STROBE system consists of a field programmable gate array (FPGA) controller attached to a multiplexor board, adaptor boards, fly arenas equipped with capacitive sensors, and lighting circuits.

The arrangement of the STROBE chambers mirrors the design of the FlyPAD, except each of the eight adaptor boards connects to two FlyPAD arenas and two lighting units instead of the original four FlyPAD arenas. These adaptor boards link the chambers to the FPGA, which is a Terasic DEV0-Nano mounted onto a custom multiplexor board with a FTDI module allowing data transfer over serial communications with a computer. The multiplexor board has eight 10-pin ports, each of which connects to an adaptor board that splits the 10-pin line into four 10-pin ports connecting to two fly arenas and two lighting circuits. The fly arena consists of two annulus shaped capacitive sensors and a CAPDAC chip (AD7150BRMZ) that the main multiplexer board communicates with to initiate and collect data (and ultimately stop collecting data). The CAPDAC interprets data from the two capacitive sensors on the fly-arena (*Itskov et al., 2014*).

The lighting circuit consists of connectors for power from an external power supply and for signaling from the FPGA controller via the intermediate components, a 617 nm light emitting diode (LUXEON Rebel LED – 127lm @ 700mA; Luxeon Star LEDs #LXM2-PH01-0060), two power resistors (TE Connectivity Passive Product SMW24R7JT) for LED current protection, and two metal oxide semiconductor field effect transistors (MOSFETs; from Infineon 634 Technologies, Neubiberg, Germany, IRLML0060TRPBF) allowing for voltage signal switching of the LEDs.

When the signal from a capacitive sensor rises during a fly sip (or other food interaction), the CAPDAC on the fly arena propagates a signal through the adaptor board via the multiplexor to the FPGA controller. The FPGA processes the capacitive sensor signal using code built atop the original VHDL code from the FlyPAD (*Itskov et al., 2018*). The STROBE VHDL code (*Chan, 2018b*) implements a running minima filter that operates in real-time to detect when a fly is feeding or otherwise interacting with the food. The filter determines the minimum signal value in the last 100 ms and compares the current signal value with this minimum. If the current signal value is greater than the minimum, and the difference between them is greater than a threshold set to exceed noise (100 units for all experiments), this is considered a rising edge and the filter will prompt the lighting activation system to activate the LED (or keep it on if it is already on). By design, this means that the control system will send a signal to deactivate the lighting upon the falling edge of the capacitance signal, or if the capacitance signal has plateaued for 100 ms, whichever comes sooner. At this point, a low signal is sent to the MOSFET which pinches off the current flowing through the lighting circuit, turning off the light. The signal to lighting response transition times are on the order of tens of milliseconds, providing a nearly instantaneous response.

The system automatically records the state of the lighting activation system (on/off) and transmits this information through USB to the PC, where it is received and interpreted by a custom end-user program (built using Qt framework in C++) which can display both the activation state and signal measured by the STROBE system in each fly arena in real-time.

All STROBE software is available for download from Github:

FPGA code: https://github.com/rcwchan/STROBE-fpga

All other code: https://github.com/rcwchan/STROBE_software/

## STROBE experiments

Flies were place in vials for three days under 'salt fed' or 'salt deprived' conditions described above. All flies were 5 – 9 days old at the time of the assay. For retinal groups, food was supplemented with all trans-Retinal at a final concentration of 1 mM (Sigma-Aldrich).

Both channels of STROBE chambers were loaded with 4 µl of 1% agar (GR64f and IR94e experiments) or 1M sucrose mixed in 1% agar (Gr66a and Ppk23$^{glut}$ experiments). Acquisition on the STROBE software was started and then single flies were transferred into each arena by mouth

aspiration. Experiments were run for 60 min, and the preference index for each fly was calculated as: (sips from Food 1 – sips from Food 2)/(sips from Food 1 + sips from Food 2).

## Statistical analysis

Statistical tests were performed using GraphPad Prism six software. Descriptions and results of each test are provided in the figure legends. Sample sizes are indicated in the figure legends. Sample sizes were determined prior to experimentation based on the variance and effect sizes seen in prior experiments of similar types. Whenever possible, all experimental conditions were run in parallel and therefore have the same or similar sample sizes. However, in some cases this was impossible, due to the concurrent availability of different genotypes or the size of the experiment. These situations account for instances where some control genotypes have very large sample sizes, since they were run in parallel with multiple experimental groups (e.g. *Figure 5A*).

All replicates were biological replicates using different flies. Data for all quantitative experiments were collected on at least three different days, and behavioral experiments were performed with flies from at least two independent crosses. Specific definitions of replicates are as follows. For calcium imaging, each data point represents the response of a single fly to the indicated stimulus. A given fly was stimulated with a specific tastant only once. For binary choice behavioral tests, each data point represents the calculated preference for a group of 10 flies. For PER, each replicate is composed of 10 independent flies tested in parallel. For STROBE experiments, each data point is the calculated preference of an individual fly over the course of the experiment.

Outliers were occasionally observed but were not removed from the datasets. For example, in *Figure 2D*, two flies had strong water responses in Gr64f sweet neurons. Although these appear to be outliers, we left them in the dataset because there was no other justification for removing them.

There were two conditions where data were excluded that were determined prior to experimentation and applied uniformly throughout. First, in calcium imaging experiments, all the data from a fly were removed if either: a) there was too much movement during stimulation to reliably quantify the response; or b) there was no response to a known, robust, positive control (rare). Second, for STROBE experiments, the data from individual flies were removed if the fly did not pass a set minimum threshold of sips (15), or the data showed hallmarks of a technical malfunction (rare).

A third condition for data exclusion arose during pilot experiments and was then applied subsequently: A subset of flies expressing GCaMP in IR94e neurons (~20%) showed a large response to water alone. These flies were removed from the analysis.

All the quantitative data used for statistical tests can be found as supplements for each figure.

## Acknowledgements

We thank Emily LeDue for preliminary PER experiments and GCaMP imaging, Simon Roome for preliminary experiments on salt deprivation, Rachael Bartlett for PCR screening of *Ppk23-Gal4*, *ΔIR76b* recombinant, Anupama Dahanukar for assistance with labellar bristle annotation, and members of the Gordon lab for comments on the manuscript. We also thank Kristin Scott, Barry Dickson, Hubert Amrein, the Bloomington Stock Center, and the Vienna *Drosophila* Resource Center for fly stocks. Carlos Ribeiro and Pavel Itskov kindly provided their original FlyPAD VHDL code, which was instrumental in developing the STROBE system. This work was funded primarily by the Canadian Institutes of Health Research (CIHR) operating grant FDN-148424, with the STROBE development funded by Natural Sciences and Engineering Research Council (NSERC) grants RGPIN-2016 – 03857 and RGPAS 492846 – 16, and with infrastructure funded by the Canadian Foundation for Innovation (CFI) grant 27290. M.D.G. is a CIHR New Investigator and a Michael Smith Foundation for Health Research Scholar.

## Additional information

### Funding

| Funder | Grant reference number | Author |
| --- | --- | --- |
| Canadian Institutes of Health Research | FDN-148424 | Michael D Gordon |

| Natural Sciences and Engineering Research Council of Canada | RGPIN-2016-03857 | Michael D Gordon |
| --- | --- | --- |
| Natural Sciences and Engineering Research Council of Canada | RGPAS-492846-16 | Michael D Gordon |

The funders had no role in study design, data collection and interpretation, or the decision to submit the work for publication.

## Author contributions

Alexandria H Jaeger, Formal analysis, Investigation, Visualization, Writing—review and editing, Carried out the labellum expression mapping and much of the GCaMP imaging of wild-type GRNs; Molly Stanley, Formal analysis, Investigation, Visualization, Writing—review and editing, Performed imaging of mutant GRNs, Contributed to wild-type GRN imaging, Contributed to behavioural testing; Zachary F Weiss, Formal analysis, Investigation, Visualization, Writing—review and editing, Carried out GRN silencing and mutant behaviour; Pierre-Yves Musso, Formal analysis, Investigation, Visualization, Writing—review and editing, Performed most of the STROBE experiments; Rachel CW Chan, Resources, Software, Methodology, Designed and built the STROBE; Han Zhang, Resources, Methodology, Designed and built the STROBE; Damian Feldman-Kiss, Formal analysis, Investigation, Identified the IR94e-Gal4 driver and performed the IR94e STROBE experiment; Michael D Gordon, Conceptualization, Formal analysis, Supervision, Funding acquisition, Investigation, Visualization, Methodology, Writing—original draft, Project administration, Writing—review and editing, Performed Ir94e labellum staining

## Author ORCIDs

Rachel CW Chan (iD) https://orcid.org/0000-0003-1009-6379
Michael D Gordon (iD) http://orcid.org/0000-0002-5440-986X

## Decision letter and Author response

Decision letter https://doi.org/10.7554/eLife.37167.027
Author response https://doi.org/10.7554/eLife.37167.028

# Additional files

## Supplementary files

• Transparent reporting form
DOI: https://doi.org/10.7554/eLife.37167.025

## Data availability

All data generated or analysed during this study are included in the manuscript and supporting files. Source data files have been provided for Figures 2-6.

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
