## [Decision Letter]

Thank you for submitting your article "A complex peripheral code for salt taste in *Drosophila*" for consideration by *eLife*. Your article has been reviewed by three peer reviewers, including Liqun Luo as the Reviewing Editor and Reviewer #1, and the evaluation has been overseen by a Senior Editor. The following individual involved in review of your submission has agreed to reveal their identity: Hubert O Amrein (Reviewer #2).

The reviewers have discussed the reviews with one another and the Reviewing Editor has drafted this decision to help you prepare a revised submission.

The four specific questions below summarized new experiments that need to be done to address reviewers' major critiques. Please see further below specific critiques from all three reviewers.

1) Ascertain the behavioral role of IR94e neurons by inactivating them with a different effector and by inactivating them together with Gr64f neurons.

2) Better define the behavioral roles of IR76b molecule and ppk23 neurons in high salt avoidance by using a preference assay where the two options contain the same amount of sugar.

3) Ascertain whether the cellular and behavioral contribution of IR76b to salt sensing acts cell autonomously in different salt-sensing neurons by restoring the expression of IR76b in these neurons.

4) Assess the role of IR25a in salt-sensing neurons whose cellular and behavioral salt responses depended on IR76b.

*Reviewer #1:*

In this study, Jaeger et al. provided a comprehensive molecular map of the gustatory receptor neurons (GRNs) in the fly labellum. They then characterized salt responses by calcium imaging of their central projections, finding that all GRN classes in the labellum respond to salt (either activated or inhibited, at specific concentration ranges). Using intersectional approaches, they found that Ppk23+ GRNs belong to two subgroups, those that are cholinergic that also express bitter receptors and respond to bitter compounds, and those that are glutamatergic and do not respond to bitter compounds; both groups are activated by high concentration of salt (high-salt). Behavioral experiments identified cell types that mediate low-salt attraction and high-salt avoidance. Interestingly, the response of glutamatergic Ppk23+ but not cholinergic Ppk23+ GRNs is modulated by whether the flies were salt-satiated or deprived. Along the way, the authors also resolved a conflict in the literature regarding the requirement of Ir76b in salt responses.

This is a comprehensive study about salt sensation in flies, highlighting the richness and complexity of this taste modality. The technical quality is also high. I am very enthusiastic in supporting its publication in *eLife*.

My only comment regards to the presentation. The Introduction contains a lot of details of what is known about the fly gustatory system prior to this study. While scholarly, it could be overwhelming for a general audience. I suggest that the authors rearrange Figure 1, to make the current Figure 1M as Figure 1A. They can then add Figure 1B in the same style as current Figure 1N, summarizing what is known about the GRN organization prior to this study. Both panels should be cited in the Introduction to help readers navigate the fly taste system. The authors should keep the current Figure 1N, which can be compared to Figure 1B to highlight what is new from this study, and serves as a summary for the current state of knowledge.

*Reviewer #2:*

In this paper, Jaeger and colleagues investigate the cellular basis of salt taste, one of the five basic taste modalities, shared by a range of different animals, including many mammals and insects. Salt is an essential nutrient component, but it can be harmful if ingested in large amounts, and hence, consumption of salt must be carefully calibrated. One of the main gatekeepers of salt intake is the taste system, and in this paper the authors use the *Drosophila* labial palp as an experimental system to dissect the cellular basis of salt taste. Overall, this is a solid paper that expands some recent reports from two labs (Montell and Kang). Like the Kang study, Jaeger and colleagues question some of the findings by the Montell laboratory, specifically the claim that IR76b is exclusively involved in low salt responses and salt attraction to substrates containing low salt. The paper substantially expands the understanding of low salt sensitivity across the spectrum of labial gustatory neurons and the study provides new insights into the cellular basis for salt taste and is therefore of considerable interest to researchers in various fields of sensory perception as well as nutrition.

A few issues need to be addressed, however, and some claims need to either be properly supported by data or they need to be toned down. In addition, some experiments to confirm the claims made in the paper need to be shown (some might have been done already), while other experiments not requiring much work would provide significant and novel insight and would increase the appeal of the study. I do think the study is of high quality, and upon revision, very suitable for publication in *eLife*.

The first issue that needs further clarification is the role of the low salt neurons. Specifically, the authors should more thoroughly investigate the role of the IR94a expressing neurons, because this is the only neuron population that responds to low salt, but not to high salt. To confirm IR94e expressing neurons are low salt neurons, it would be helpful to include a couple of additional concentrations between 50mM and 250mM. In addition, while their experiment to disable the function of these neurons with Kir2.1 shows no significant phenotype in low salt attraction, the authors should try other strategies, such as tetanus toxin mediated inactivation or cell killing with diphtheria toxin. A role in low salt attraction of these neurons is consistent with the observation that Kir2.1 mediated suppression of sweet neurons leaves flies still attracted to low salt, so a combination of IR94a-GAL4 and Gr64f-GAL4 driving UAS-Kir2.1 or one of the other effectors might eliminate low salt attraction completely. Also, do Ir94a neurons express Ir76b? Even if they do, Ir76b shows variable levels of expression in a number of neurons (see Chen and Amrein, 2017; Ahn et al., *eLife*2017), and the fact that Ir76b-GAL4, UAS-Kir2.1 flies show remaining low salt attraction might be due to this.

Second, the authors show no rescue experiments for the IR76b mutant phenotype, which is a common test to prove casualty of phenotype, especially in *Drosophila*. In fact, they could use various drivers that define subpopulation of neurons to test the requirement of Ir76b in these subsets.

Third, in Figure 2 they show that Gr64f expressing GRNs respond to high salt, not only low salt. It is likely that the modulation phenotype of the internal state (Figure 5) is not only mediated by ppk23glut and bitter GRNs, but also by sweet GRNs. That can easily be tested.

And last, the authors' data question aspects of the role of Ir76b in salt taste, previously reported by Zhang et al., 2013, and probably justly so. Zhang also claimed that Ir25a plays no role in salt taste. That claim seems odd, because all Ir based receptors, either in the olfactory or taste system, appear to include either Ir8a or Ir25a. To complete the comparison, the authors should test whether or not Ir25a plays a role in salt taste in the neurons in which salt responses are affected by Ir76b mutations. I would not ask them to do more than that, but given the general mutual dependence between Ir25a and Ir76b in taste, it would be important the check for that.

*Reviewer #3:*

In this study, Jaeger and colleagues did a nice characterization of salt responses of labellum taste neurons in flies. They first assembled tools that target different groups of labellar neurons; they then characterized the cellular responses of these neurons to different concentrations of salts. Finally, they determined the contribution of different salt-responsive neurons to salt-induced behaviors. Based on their results, the authors concluded that flies use distinct groups of neurons to sense and behaviorally respond to high vs. low salts. Specifically, detection and behavioral attraction for low salt is mediated by sweet (Gr64f) neurons whereas detection and avoidance of high salt is mediated by two other groups: one is defined by its expression of vGlut and ppk23 while another defined by their expression of Gr66a and ppk23. Moreover, the authors showed that both the Gr64f and ppk23/vGlut group rely on IR76b to sense salts whereas the ppk23/Gr66a group does not. Lastly, while the ppk23/Gr66a group can drive avoidance of high salt independent of salt intake, the ppk23/vGlut group is less able to drive high salt avoidance when animals have been salt deprived.

In general, I find the experiments were well executed and the messages very interesting. But I do have a few concerns about the claim that Gr64f and the ppk23/vGlut neurons use IR76b for salt-sensing.

1) Why is it that silencing the Gr64f neurons caused an indifference towards low salt but removing IR76b caused strong avoidance of low salt? What causes such strong avoidance in IR76b mutants and why is it not active in Gr64f-silenced animals?

2) Why is it that silencing ppk23/vGlut and ppk23/Gr66a neurons caused a clear reduction of avoidance of high salt (in a 250mM NaCl + 25mM sucrose vs. 0mM NaCl + 5mM sucrose preference test) but removing IR76b caused a strong preference for high salt? What is the source of such strong preference? From the sweet neurons? Also, how come the Gr66a/ppk23 neurons were not able to counter some of the attraction for high salt?

3) Related to point (2), I think it is important to use the same sucrose concentration on both options when asking the animals choose between 250 mM NaCl vs. no salt so that it will be easier to interpret the results. If the authors' proposal is correct, then the expectation is that IR76b mutants should show avoidance of high salt (mediated by ppk23/Gr66a neurons) whereas silencing all ppk23 neurons should drive preference for high salt (mediated by Gr64f neurons).

4) The requirement of IR76b in sweet and ppk23/vGlut neurons for salt sensing would be more convincing if one can show that IR76b acts cell autonomously in them to influence Ca^2+^ and behavioral responses. This could be done by restoring IR76b to these neurons or removing IR76 in them by RNAi. While the GCaMP recording was done nicely, I wonder whether it is possible that such responses may also reflect changes in other neurons that then influence the neurons of interest via presynaptic modulation.

---

## [Author Response]

The four specific questions below summarized new experiments that need to be done to address reviewers' major critiques. Please see further below specific critiques from all three reviewers.1) Ascertain the behavioral role of IR94e neurons by inactivating them with a different effector and by inactivating them together with Gr64f neurons.

We have completed these experiments. Inactivating IR94e and Gr64f together with Kir2.1 produced a loss of low salt attraction that was indistinguishable from Gr64f inactivation alone (Figure 5A). However, to our surprise, inactivation of IR94e alone with tetanus toxin (TNT) partially reduced low salt attraction (Figure 5—figure supplement 1). We don’t know for sure why there is a difference between these effectors. It is possible that TNT produces stronger inhibition of GRNs than Kir2.1, or that the temporal control with Gal80^ts^ that we employed with Kir2.1, but not with TNT, resulted in weaker effects. Between these data and the result that Gr64f neurons produce strong attraction upon optogenetic activation, but IR94e neuron activation produced no effect, we conclude that low salt attraction is primarily mediated by Gr64f GRNs, with additional contribution from IR94e neurons.

2) Better define the behavioral roles of IR76b molecule and ppk23 neurons in high salt avoidance by using a preference assay where the two options contain the same amount of sugar.

We completely understand the desire for such experiments, but, unfortunately, they don’t yield informative results. If the two sugar concentrations are equal and one is mixed with an aversive concentration of salt, the flies, no matter the genotype, uniformly select the salt-free option. This is true for all the sugar and salt concentrations we tested. It is also true for bitter compounds, which is why essentially all published binary taste preference tests for aversive compounds involve mixing the aversive tastant with a higher concentration of sugar than what is present in the other option (for examples, see Weiss et al., 2011 or Lee et al., 2018). Without this extra incentive to consume the option laced with the aversive compound, even a small amount of residual salt/bitter detection is enough to mediate almost complete avoidance. This residual aversive activity could be from many sources, including:

1) Detection by another set of gustatory neurons – this is particularly relevant in the case of Ppk23^glut^ and Gr66a, because silencing either alone leaves the other intact, which is sufficient to drive avoidance.

2) Incomplete inactivation of the specific neurons targeted – this could be from incomplete silencing by the effector, or from the Gal4 not expressing in the complete set of relevant neurons.

3) The aversive compound causing interference with sugar detection – this is particularly relevant for bitter compounds, but may also be in play with salts (see Lee et al., 2018 for suppression of sugar activity by calcium).

4) Taste-independent homeostatic mechanisms to prevent excess salt consumption.

Thus, when both options contain the same concentration of sugar, a significant reduction in avoidance can only be observed when there is near complete loss of aversive activity. Given the multiple mechanisms that mediate aversion, this is not a practical scenario for uncovering the roles of specific mechanisms in behavior. The unequal sugar concentrations allow us to sensitize the assay so that partial loss of aversive input can still be seen as a significant change in preference.

Having said that, we certainly understand the desire for a more intuitive (or at least different) readout of high salt avoidance. Therefore, we extended our PER analyses to include Ppk23^glut^ and Gr66a inactivation under both salt fed and salt deprived conditions. These data, which are shown in Figure 5—figure supplement 2, support the conclusions from the binary choice assays. We see a major role for Ppk23^glut^ GRNs in PER suppression by high salt in flies fed a salt-containing diet, but no impact in flies that have been salt deprived. Conversely, we saw only minor effects from silencing Gr66a neurons, and these effects were more pronounced in the salt deprived flies.

3) Ascertain whether the cellular and behavioral contribution of IR76b to salt sensing acts cell autonomously in different salt-sensing neurons by restoring the expression of IR76b in these neurons.

We have done these experiments. The data from rescuing IR76b function in both calcium imaging and behavior are presented in Figure 4. All the data support cell autonomous roles for IR76b in salt detection by each GRN type.

4) Assess the role of IR25a in salt-sensing neurons whose cellular and behavioral salt responses depended on IR76b.

We have performed calcium imaging on all three major salt-responsive GRN types in *IR25a* mutants and cell specific rescues. These data, presented in Figure 4—figure supplement 2, reveal requirements for *IR25a* in salt detection that are very similar to what we saw for *IR76b*. The most straightforward interpretation of these results is that IR76b and IR25a act in a complex to mediate both low and high salt detection.

Reviewer #1:[…] This is a comprehensive study about salt sensation in flies, highlighting the richness and complexity of this taste modality. The technical quality is also high. I am very enthusiastic in supporting its publication in eLife.My only comment regards to the presentation. The Introduction contains a lot of details of what is known about the fly gustatory system prior to this study. While scholarly, it could be overwhelming for a general audience. I suggest that the authors rearrange Figure 1, to make the current Figure 1M as Figure 1A. They can then add Figure 1B in the same style as current Figure 1N, summarizing what is known about the GRN organization prior to this study. Both panels should be cited in the Introduction to help readers navigate the fly taste system. The authors should keep the current Figure 1N, which can be compared to Figure 1B to highlight what is new from this study, and serves as a summary for the current state of knowledge.

We appreciate the enthusiasm, and also the excellent suggestion for the Introduction – the details can be hard to keep straight, even for us! We have modified Figure 1 accordingly and referenced it in the Introduction.

Reviewer #2:[…] A few issues need to be addressed, however, and some claims need to either be properly supported by data or they need to be toned down. In addition, some experiments to confirm the claims made in the paper need to be shown (some might have been done already), while other experiments not requiring much work would provide significant and novel insight and would increase the appeal of the study. I do think the study is of high quality, and upon revision, very suitable for publication in eLife.The first issue that needs further clarification is the role of the low salt neurons. Specifically, the authors should more thoroughly investigate the role of the IR94a expressing neurons, because this is the only neuron population that responds to low salt, but not to high salt. To confirm IR94e expressing neurons are low salt neurons, it would be helpful to include a couple of additional concentrations between 50mM and 250mM.

We have now added calcium imaging data of the IR94e population stimulated with 100 mM concentrations of a variety of salts. These data, shown in Figure 2—figure supplement 1, confirm mild activation by low concentrations of sodium salts. The fact that these responses are sodium-specific, even at low concentrations, fits nicely with the annotation of IR94e GRNs as a ‘low salt’ cell type.

In addition, while their experiment to disable the function of these neurons with Kir2.1 shows no significant phenotype in low salt attraction, the authors should try other strategies, such as tetanus toxin mediated inactivation or cell killing with diphtheria toxin. A role in low salt attraction of these neurons is consistent with the observation that Kir2.1 mediated suppression of sweet neurons leaves flies still attracted to low salt, so a combination of IR94a-GAL4 and Gr64f-GAL4 driving UAS-Kir2.1 or one of the other effectors might eliminate low salt attraction completely.

As described above (Editor’s point #1), we have done the suggested experiments and they revealed a likely minor role for IR94e cells in low salt attraction. We appreciate the suggestions.

Also, do Ir94a neurons express Ir76b? Even if they do, Ir76b shows variable levels of expression in a number of neurons (see Chen and Amrein, 2017; Ahn et al., 2017), and the fact that Ir76b-GAL4, UAS-Kir2.1 flies show remaining low salt attraction might be due to this.

We don’t know for sure whether IR94e neurons express *IR76b*. We only have a Gal4 line for IR94e so we could not assess overlap with *IR76b-Gal4* as we did for the other populations. We could have used *IR76b-QF* instead, but because there is substantial variability between different IR76b reporters, we didn’t think this would be particularly informative. We also piloted IR94e GCaMP imaging in *IR76b* mutants, but the calcium responses in these neurons are so weak and variable that we could not make confident conclusions about their dependence on IR76b. It’s also worth noting here that, although there is a lot of interest from the reviewers about the receptor requirements in various populations, it was never our intention to make this a primary focus of the paper. Our focus is on the coding of salt taste across different populations, and we delved into *IR76b* (and now *IR25a*) requirements only as an attempt to reconcile our results with prior studies.

Second, the authors show no rescue experiments for the IR76b mutant phenotype, which is a common test to prove casualty of phenotype, especially in Drosophila. In fact, they could use various drivers that define subpopulation of neurons to test the requirement of Ir76b in these subsets.

This has been done, as described above in Editor’s point #3.

Third, in Figure 2 they show that Gr64f expressing GRNs respond to high salt, not only low salt. It is likely that the modulation phenotype of the internal state (Figure 5) is not only mediated by ppk23glut and bitter GRNs, but also by sweet GRNs. That can easily be tested.

Although we agree that Gr64f GRNs may play a role in the observed modulation, especially at low salt concentrations, this is actually not trivial to test. Testing the role of Gr64f neurons in the high salt avoidance assay is fraught because silencing Gr64f neurons will affect their sugar responses and therefore their preference in the assay. On the other hand, we have done the low salt attraction assay in salt fed and salt deprived conditions, and as one would expect, we see attraction only in salt deprived flies (data not shown). In this case, silencing Gr64f neurons produces a mild, but significant, avoidance of 50 mM salt when the flies are salt fed. This suggests that Gr64f neurons are still active in driving salt attraction in salt fed flies (as they are in salt deprived flies), but that salt feeding either a) introduces a counteracting negative drive from some other population, or b) substantially reduces the attraction mediated by Gr64f neurons. Differentiating between these possibilities would be difficult, given the mild nature of the phenotypes, and we felt pursuing this direction adds complication to the story without necessarily much added insight. Therefore, we decided to leave this data out of the paper.

And last, the authors' data question aspects of the role of Ir76b in salt taste, previously reported by Zhang et al., 2013, and probably justly so. Zhang also claimed that Ir25a plays no role in salt taste. That claim seems odd, because all Ir based receptors, either in the olfactory or taste system, appear to include either Ir8a or Ir25a. To complete the comparison, the authors should test whether or not Ir25a plays a role in salt taste in the neurons in which salt responses are affected by Ir76b mutations. I would not ask them to do more than that, but given the general mutual dependence between Ir25a and Ir76b in taste, it would be important the check for that.

We have done these experiments, and as you suspected, IR25a does indeed play a role in salt taste. This is addressed in Editor’s point #4 above.

Reviewer #3:In this study, Jaeger and colleagues did a nice characterization of salt responses of labellum taste neurons in flies. They first assembled tools that target different groups of labellar neurons; they then characterized the cellular responses of these neurons to different concentrations of salts. Finally, they determined the contribution of different salt-responsive neurons to salt-induced behaviors. Based on their results, the authors concluded that flies use distinct groups of neurons to sense and behaviorally respond to high vs. low salts. Specifically, detection and behavioral attraction for low salt is mediated by sweet (Gr64f) neurons whereas detection and avoidance of high salt is mediated by two other groups: one is defined by its expression of vGlut and ppk23 while another defined by their expression of Gr66a and ppk23. Moreover, the authors showed that both the Gr64f and ppk23/vGlut group rely on IR76b to sense salts whereas the ppk23/Gr66a group does not. Lastly, while the ppk23/Gr66a group can drive avoidance of high salt independent of salt intake, the ppk23/vGlut group is less able to drive high salt avoidance when animals have been salt deprived. In general, I find the experiments were well executed and the messages very interesting. But I do have a few concerns about the claim that Gr64f and the ppk23/vGlut neurons use IR76b for salt-sensing.1) Why is it that silencing the Gr64f neurons caused an indifference towards low salt but removing IR76b caused strong avoidance of low salt? What causes such strong avoidance in IR76b mutants and why is it not active in Gr64f-silenced animals?

This puzzled us as well. However, when we repeated the IR76b mutant behavior in two different control populations for the rescue experiment, we did not see the same aversion (Figure 4D). In fact, their behavior was much more similar to Gr64f silencing. We don’t know why these results differ from the original experiment, but since the new experiments involve more independent replicates across two populations, we decided to trust them more than the old ones. In any case, the residual attraction seen after Gr64f silencing is likely to be from IR94e neurons, based on double silencing of these two populations with TNT (see Editor’s point #1 above).

2) Why is it that silencing ppk23/vGlut and ppk23/Gr66a neurons caused a clear reduction of avoidance of high salt (in a 250mM NaCl + 25mM sucrose vs. 0mM NaCl + 5mM sucrose preference test) but removing IR76b caused a strong preference for high salt? What is the source of such strong preference? From the sweet neurons? Also, how come the Gr66a/ppk23 neurons were not able to counter some of the attraction for high salt?

As you point out below, the answer is related to your point #3. The attraction is not likely attraction to the high salt, but rather attraction to the higher sugar concentration, coupled with reduced avoidance of high salt. Presumably, what we are seeing with silencing of either Ppk23^glut^ or Gr66a populations is a partial loss of high salt avoidance, because in each case the other population is there to mediate some avoidance. IR76b mutations cause a more extensive loss of high salt activity, because there is a (near) complete loss in Ppk23^glut^ activation, and a partial loss of Gr66a activation. The fact that the preference is near 1 doesn’t imply that there is no residual avoidance from Gr66a, since it may be overridden by the higher sugar concentration. Additionally, the silencing phenotypes may be incomplete because Kir2.1 is not eliminating all neuron activity, whereas the *IR76b* mutants are a complete loss of function. Finally, it’s important to keep in mind that the *IR76b* mutants may have other defects that are unaccounted for – either function in other tissues (even other gustatory organs that may not have exactly the same logic as the labellum) or changes in internal state that are driven by the fact that these mutants have been defective for many chemosensory functions their entire lives.

3) Related to point (2), I think it is important to use the same sucrose concentration on both options when asking the animals choose between 250 mM NaCl vs. no salt so that it will be easier to interpret the results. If the authors' proposal is correct, then the expectation is that IR76b mutants should show avoidance of high salt (mediated by ppk23/Gr66a neurons) whereas silencing all ppk23 neurons should drive preference for high salt (mediated by Gr64f neurons).

Please see our response to Editor’s point #2.

4) The requirement of IR76b in sweet and ppk23/vGlut neurons for salt sensing would be more convincing if one can show thatIR76b acts cell autonomously in them to influence Ca^2+^ and behavioral responses. This could be done by restoring IR76b to these neurons or removing IR76 in them by RNAi. While the GCaMP recording was done nicely, I wonder whether it is possible that such responses may also reflect changes in other neurons that then influence the neurons of interest via presynaptic modulation.

We have done the cell-specific rescues in Ppk23, Gr66a, and Gr64f populations, for both calcium imaging and behavior (Figure 4; also see Editor’s point #3). These data all support cell autonomous IR76b function in salt sensing.